# Tighter Expected Generalization Error Bounds via Wasserstein Distance

**Borja Rodríguez-Gálvez**
KTH Royal Institute of Technology
Stockholm, Sweden
borjarg@kth.se

**Germán Bassi**
Ericsson Research
Stockholm, Sweden
german.bassi@ericsson.com

**Ragnar Thobaben**
KTH Royal Institute of Technology
Stockholm, Sweden
ragnart@kth.se

**Mikael Skoglund**
KTH Royal Institute of Technology
Stockholm, Sweden
skoglund@kth.se

## Abstract

This work presents several expected generalization error bounds based on the Wasserstein distance. More specifically, it introduces full-dataset, single-letter, and random-subset bounds, and their analogues in the randomized subsample setting from Steinke and Zakynthinou [1]. Moreover, when the loss function is bounded and the geometry of the space is ignored by the choice of the metric in the Wasserstein distance, these bounds recover from below (and thus, are tighter than) current bounds based on the relative entropy. In particular, they generate new, non-vacuous bounds based on the relative entropy. Therefore, these results can be seen as a bridge between works that account for the geometry of the hypothesis space and those based on the relative entropy, which is agnostic to such geometry. Furthermore, it is shown how to produce various new bounds based on different information measures (e.g., the lautum information or several $f$-divergences) based on these bounds and how to derive similar bounds with respect to the backward channel using the presented proof techniques.

## 1   Introduction

A *learning algorithm* is a mechanism that takes a *dataset $s = (z_1, \ldots, z_n)$* of $n$ samples $z_i \in \mathcal{Z}$ taken i.i.d. from a distribution $P_Z$ as an input, and produces a hypothesis $w \in \mathcal{W}$ by means of the conditional probability distribution $P_{W|S}$.

The ability of a hypothesis $w$ to characterize a sample $z$ is described by the loss function $\ell(w, z) \in \mathbb{R}$. More precisely, a hypothesis $w$ describes well the samples from a population $P_Z$ when its *population risk*, i.e., $\mathscr{L}_{P_Z}(w) \triangleq \mathbb{E}[\ell(w, Z)]$, is low. However, the distribution $P_Z$ is often not available and the *empirical risk* on the dataset $s$, i.e., $\mathscr{L}_s(w) \triangleq \frac{1}{n} \sum_{i=1}^{n} \ell(w, z_i)$, is considered as a proxy. Therefore, it is of interest to study the discrepancy between the population and empirical risks, which is defined as the *generalization error*:

$$\text{gen}(w, s) \triangleq \mathscr{L}_{P_Z}(w) - \mathscr{L}_s(w).$$

Classical approaches bound the generalization error in expectation and in probability (PAC Bayes) either by studying the complexity and the geometry of the hypothesis' space $\mathcal{W}$ or by exploring properties of the learning algorithm itself; see, e.g., [2, 3] for an overview.

More recently, the relationship (or amount of information) between the generated hypothesis and the training dataset has been used as an indicator of the generalization performance. In [4], based on

35th Conference on Neural Information Processing Systems (NeurIPS 2021).

[5], it is shown that the expected generalization error, i.e., $\overline{\text{gen}}(W, S) \triangleq \mathbb{E}[\text{gen}(W, S)]$, is bounded from above by a function that depends on the mutual information between the hypothesis $W$ and the dataset $S$ with which it is trained, i.e., $I(W; S)$. However, this bound becomes vacuous when $I(W; S) \to \infty$, which occurs for example when $W$ and $S$ are separately continuous and $W$ is a deterministic function of $S$. To address this issue, it is shown in [6] that the generalization error is also bounded by a function on the dependency between the hypothesis and individual samples, $I(W; Z_i)$, which is usually finite due to the smoothing effect of marginalization. Following this line of work, in [7], the authors present data-dependent bounds based on the relationship between the hypothesis and random subsets of the data, i.e., $D_{\text{KL}}(P_{W|s} \| P_{W|s_{j^c}})$ where $j \subseteq [n]$.

After that, a more structured setting is introduced in [1], studying instead the relationship between the hypothesis and the *identity* of the samples. The authors consider a super-sample of $2n$ i.i.d. instances $\tilde{z}_i$ from $P_Z$, i.e., $\tilde{s} = (\tilde{z}_1, \ldots, \tilde{z}_{2n})$. This super sample is used to construct the dataset $s$ by choosing between the samples $\tilde{z}_i$ and $\tilde{z}_{i+n}$ using a Bernoulli random variable $U_i$ with probability $\frac{1}{2}$, i.e., $z_i = \tilde{z}_{i+u_i n}$. In this paper, the two settings are referred to as the *standard* and *randomized-subsample* settings.[1] In the randomized-subsample setting, the *empirical generalization error* is defined as the difference between the empirical risk on the samples from $\tilde{s}$ not used to obtain the hypothesis, i.e., $\bar{s} = \tilde{s} \setminus s$, and the empirical risk on the dataset $s$, i.e.,

$$\widehat{\text{gen}}(w, \tilde{s}, u) \triangleq \mathscr{L}_{\bar{s}}(w) - \mathscr{L}_s(w) = \frac{1}{n} \sum_{i=1}^{n} \left( \ell(w, \tilde{z}_{i+(1-u_i)n}) - \ell(w, \tilde{z}_{i+u_i n}) \right),$$

where $u$ is the sequence of $n$ i.i.d. Bernoulli trial outcomes $u_i$. The expected value of the empirical and the (standard) generalization errors coincide, i.e., $\mathbb{E}[\widehat{\text{gen}}(W, \tilde{S}, U)] = \overline{\text{gen}}(W, S)$. Also, the expected generalization error is controlled by the conditional mutual information between the hypothesis $W$ and the Bernoulli trials $U$, given the super sample $\tilde{S}$ [1], i.e., $I(W; U|\tilde{S})$, by the individual conditional mutual information [9], i.e., $I(W; U_i|\tilde{Z}_i, \tilde{Z}_{i+n})$, and by the "disintegrated" mutual information with a subset $U_J$ of the Bernoulli trials [10], i.e., $D_{\text{KL}}(P_{W|\tilde{S}, U} \| P_{W|\tilde{S}, U_{J^c}})$. A highlight of this setting is that these conditional notions of information are always finite [1] and smaller than their "unconditional" counterparts [10], e.g., $I(W; U|\tilde{S}) \leq I(W; S)$ and $I(W; U|\tilde{S}) \leq n \log(2)$.

Some steps towards unifying these results are taken in [8], where the authors develop a framework that makes it possible to recover the expected generalization error bounds based on the mutual information $I(W; S)$ and the conditional mutual information $I(W; U|\tilde{S})$. Then, the aforementioned framework is further exploited in [9] to recover the single-sample and the random-subsets bounds, which are based on $I(W; Z_i)$, $D_{\text{KL}}(P_{W|S} \| P_{W|S_{J^c}})$, and $D_{\text{KL}}(P_{W|\tilde{S}, U} \| P_{W|\tilde{S}, U_{J^c}})$, and to generate new individual conditional mutual information bounds, i.e., $I(W; U_i|\tilde{Z}_i, \tilde{Z}_{i+n})$. Finally, in [11, 12], other systematic ways to recover some of the said bounds and obtain similar new ones are studied.

In parallel, there were some attempts to bridge the gap between employing the geometry and complexity of the hypothesis space and the relationship between the hypothesis and the training samples. In [13], the authors bound $\overline{\text{gen}}(W, S)$ with a function of weighted dependencies between the dataset and increasingly finer quantizations of the hypothesis, i.e., $\{2^{-k/2} I([W]_k; S)\}_k$, which can be finite even if $I(W; S) \to \infty$. This result stems from a clever usage of the chaining technique [14, Theorem 5.24], and a comparison with this kind of approaches is given in Appendix A. Later, in [15] and [16], it is shown that the expected generalization error is bounded from above by a function of the Wasserstein distance between the hypothesis distribution after observing the dataset $P_{W|S}$ and its prior $P_W$, i.e., $\mathbb{W}_p(P_{W|S}, P_W)$, and by a function of the Wasserstein distance between the hypothesis distribution after observing a single sample $P_{W|Z_i}$ and its prior $P_W$, i.e., $\mathbb{W}_p(P_{W|Z_i}, P_W)$, which are finite when a suitable metric is chosen but difficult to evaluate. Concurrently, in [17] it is shown that a similar result holds, if the metric is the Minkowski distance, for the distribution of the data $P_S$ and the backward channel $P_{S|W}$, i.e., $\mathbb{W}_{p, \|\cdot\|}^p(P_{S|W}, P_S)$.

The main contributions of this paper are the following:

- It introduces new, tighter single letter and random-subset Wasserstein distance bounds for the standard and randomized-subsample settings (Theorems 1, 2, 3, and 4).

- It shows that when the loss is bounded and the geometry of the space is ignored, these bounds recover from below (and thus are tighter than) the current relative entropy and mutual

---

[1] In [8] the latter is called the random-subset setting. However, this may cause confusion with the random-subset bounds in the present work.

information bounds on both the standard and randomized-subsample settings. In fact, they are also tighter when the loss is additionally subgaussian or under certain milder conditions on the geometry. However, these results are deferred to Appendix B to expose the main ideas more clearly. Moreover, Corollaries 1 and 2 overcome the issue of potentially vacuous relative entropy bounds on the standard setting.

- It introduces new bounds based on the backward channel, which are analogous to those based on the forward channel and more general than previous results in [17].

- It shows how to generate new bounds based on a variety of information measures, e.g., the lautum information or several $f$-divergences like the Hellinger distance or the $\chi^2$-divergence, thus making the characterization of the generalization more flexible.

## 2 Preliminaries

### 2.1 Notation

Random variables $X$ are written in capital letters, their realizations $x$ in lower-case letters, their set of outcomes $\mathcal{X}$ in calligraphic letters, and their Borel $\sigma$-algebras $\mathfrak{X}$ in script-style letters. Moreover, the probability distribution of a random variable $X$ is written as $P_X : \mathfrak{X} \to [0,1]$. Hence, the random variable $X$ or the probability distribution $P_X$ induce the probability space $(\mathcal{X}, \mathfrak{X}, P_X)$. When more than one random variable is considered, e.g., $X$ and $Y$, their joint distribution is written as $P_{X,Y} : \mathfrak{X} \otimes \mathcal{Y} \to [0,1]$ and their product distribution as $P_X \otimes P_Y : \mathfrak{X} \otimes \mathcal{Y} \to [0,1]$. Moreover, the conditional probability distribution of $Y$ given $X$ is written as $P_{Y|X} : \mathcal{Y} \otimes \mathcal{X} \to [0,1]$ and defines a probability distribution $P_{Y|X=x}$ (or $P_{Y|x}$ for brevity) over $\mathcal{Y}$ for each element $x \in \mathcal{X}$. Finally, there is an abuse of notation writing $P_{X,Y} = P_{Y|X} \times P_X$ since $P_{X,Y}(B) = \int \left( \int \chi_B((x,y)) dP_{Y|X=x}(y) \right) dP_X(x)$ for all $B \in \mathfrak{X} \otimes \mathcal{Y}$, where $\chi_B$ is the characteristic function of the set $B$. The natural logarithm is log.

### 2.2 Necessary definitions, remarks, claims, and lemmas

**Definition 1.** *Let $\rho : \mathcal{X} \times \mathcal{X} \to \mathbb{R}_+$ be a metric. A space $(\mathcal{X}, \rho)$ is Polish if it is complete and separable. Throughout it is assumed that all Polish spaces $(\mathcal{X}, \rho)$ are equipped with the Borel $\sigma$-algebra $\mathfrak{X}$ generated by $\rho$. When there is no ambiguity, both the metric space $(\mathcal{X}, \rho)$ and the generated measurable space $(\mathcal{X}, \mathfrak{X})$ are written as $\mathcal{X}$.*

**Definition 2.** *Let $(\mathcal{X}, \rho)$ be a Polish metric space and let $p \in [1, \infty)$. Then, the* Wasserstein distance *of order $p$ between two probability distributions $P$ and $Q$ on $\mathcal{X}$ is*

$$\mathbb{W}_p(P,Q) \triangleq \left( \inf_{R \in \Pi(P,Q)} \int_{\mathcal{X} \times \mathcal{X}} \rho(x,y)^p dR(x,y) \right)^{1/p},$$

*where $\Pi(P,Q)$ is the set of all couplings $R$ of $P$ and $Q$, i.e., all joint distributions on $\mathcal{X} \times \mathcal{X}$ with marginals $P$ and $Q$, that is, $P(B) = R(B, \mathcal{X})$ and $Q(B) = R(\mathcal{X}, B)$ for all $B \in \mathfrak{X}$.*

**Remark 1.** *Hölder's inequality implies that $\mathbb{W}_p \leq \mathbb{W}_q$ for all $p \leq q$ [18, Remark 6.6]. Hence, since this work is centered on upper bounds the focus is on $\mathbb{W} \triangleq \mathbb{W}_1$.*

**Definition 3.** *A function $f : \mathcal{X} \to \mathbb{R}$ is said to be $L$-Lipschitz under the metric $\rho$, or simply $f \in L\text{-Lip}(\rho)$, if $|f(x) - f(y)| \leq L\rho(x,y)$ for all $x, y \in \mathcal{X}$.*

**Lemma 1** (Kantorovich-Rubinstein duality [18, Remark 6.5])**.** *Let $\mathcal{P}_1(\mathcal{X})$ be the space of probability distributions on $\mathcal{X}$ with a finite first moment. Then, for any two distributions $P$ and $Q$ in $\mathcal{P}_1(\mathcal{X})$*

$$\mathbb{W}(P,Q) = \sup_{f \in 1\text{-Lip}(\rho)} \left\{ \int_{\mathcal{X}} f(x) dP(x) - \int_{\mathcal{X}} f(x) dQ(x) \right\}. \qquad \text{(KR duality)}$$

**Definition 4.** *The total variation between two probability distributions $P$ and $Q$ on $\mathcal{X}$ is*

$$\mathtt{TV}(P,Q) \triangleq \sup_{A \in \mathfrak{X}} \left\{ P(A) - Q(A) \right\}.$$

**Definition 5.** *The discrete metric is $\rho_H(x,y) \triangleq \mathbb{1}[x \neq y]$, where $\mathbb{1}$ is the indicator function.*

**Remark 2.** *A bounded function $f : \mathcal{X} \to [a,b]$ is $(b-a)$-Lipschitz under the discrete metric $\rho_H$.*

**Remark 3.** *The Wasserstein distance of order 1 is dominated by the total variation. For instance, if $P$ and $Q$ are two distributions on $\mathcal{X}$ then $\mathbb{W}(P,Q) \leq d_\rho(\mathcal{X})\mathtt{TV}(P,Q)$, where $d_\rho(\mathcal{X})$ is the diameter of $\mathcal{X}$. In particular, when the discrete metric is considered $\mathbb{W}(P,Q) = \mathtt{TV}(P,Q)$ [18, Theorem 6.15].*

**Lemma 2** (Pinsker's and Bretagnolle–Huber's (BH) inequalities). *Let $P$ and $Q$ be two probability distributions on $\mathcal{X}$ and define $\Psi(x) \triangleq \sqrt{\min\{x/2, 1 - \exp(-x)\}}$, then [19, Theorem 6.5] and [20, Proof of Lemma 2.1] state that*

$$\mathrm{TV}(P, Q) \leq \Psi(D_{\mathrm{KL}}(P \,\|\, Q)).$$

## 3  Expected Generalization Error Bounds

This section presents our main results. First, in §3.1 and §3.2, single-letter and random-subset bounds based on the Wasserstein distance are introduced for the studied settings. These subsections also show how these bounds are tighter than current bounds based on the Wasserstein distance and the relative entropy. Moreover, an example where these bounds outperform current bounds is provided. Then, in §3.3 it is shown how to obtain analogous bounds to those in §3.1 and §3.2 for the backward channel. Finally, §3.4 shows how the presented results lead to a rich set of new bounds based on different information measures. All complete proofs and technical details are deferred to the appendix.

### 3.1  Standard setting

In [15, Theorem 2], the authors show that the expected generalization error is bounded from above by the Wasserstein distance between the forward channel distribution $P_{W|S}$ and the marginal distribution of the hypothesis $P_W$. More specifically, when the loss function $\ell$ is $L$-Lipschitz under a metric $\rho$ for all $z \in \mathcal{Z}$ and the hypothesis space $\mathcal{W}$ is Polish, then

$$\left|\overline{\mathrm{gen}}(W, S)\right| \leq L\mathbb{E}\left[\mathbb{W}(P_{W|S}, P_W)\right] = L \int_{\mathcal{Z}^n} \mathbb{W}(P_{W|S=s}, P_W) dP_Z^{\otimes n}(s). \tag{1}$$

This bound considers both the geometry of the hypothesis space by means of the metric $\rho$ and the dependence between the hypothesis and the dataset via the discrepancy between the forward channel $P_{W|S}$ and the marginal $P_W$. Nonetheless, it is not clear how it relates with other results agnostic to the geometry of the space. For instance, when the loss function $\ell$ is bounded in $[a, b]$, if the geometry is ignored (i.e., the discrete metric is considered), then

$$\left|\overline{\mathrm{gen}}(W, S)\right| \leq (b - a)\mathbb{E}\left[\mathrm{TV}(P_{W|S}, P_W)\right] \leq (b - a)\Psi(I(W; S)),$$

where the inequalities follow from Remark 2, Lemma 2, and Jensen's inequality (note $\Psi(x)$, defined in Lemma 2, is concave on $x$). This result compares negatively with other results employing the mutual information, e.g., [4, Theorem 1], where the bound has a decaying factor of $1/\sqrt{n}$.

Nonetheless, it is possible to find a single-letter version of [15, Theorem 2] using a similar strategy to [6, Proposition 1] and [9, Propositions 1 and 3], which generalizes [16, Theorem 1] to algorithms that may consider the ordering of the samples. More concretely, the expected generalization error is controlled by a function of the Wasserstein distance of the hypothesis' distribution before and after observing a *single sample $Z_i$*, i.e., $\mathbb{W}(P_{W|Z_i}, P_W)$.

**Theorem 1.** *Suppose that the loss function $\ell$ is $L$-Lipschitz for all $z \in \mathcal{Z}$ and that the hypothesis space $\mathcal{W}$ is Polish. Then,*

$$\left|\overline{\mathrm{gen}}(W, S)\right| \leq \frac{L}{n} \sum_{i=1}^{n} \mathbb{E}\left[\mathbb{W}(P_{W|Z_i}, P_W)\right].$$

Moreover, when the loss function is bounded and the geometry of the space is ignored by considering the discrete metric, this single-letter result can improve upon current relative entropy and mutual information bounds.

**Corollary 1.** *Under the conditions of Theorem 1, if the loss $\ell$ is bounded in $[a, b]$, then*

$$\left|\overline{\mathrm{gen}}(W, S)\right| \leq \frac{b - a}{n} \sum_{i=1}^{n} \mathbb{E}\left[\mathrm{TV}(P_{W|Z_i}, P_W)\right] \leq \frac{b - a}{n} \sum_{i=1}^{n} \mathbb{E}\left[\Psi\left(D_{\mathrm{KL}}(P_{W|Z_i} \,\|\, P_W)\right)\right]$$

Corollary 1 improves upon [6, Proposition 1] in two different ways. First, it pulls the expectation with respect to the samples $P_{Z_i}$ outside of the concave square root, thus strengthening that result via Jensen's inequality. Second, the addition of the BH inequality ensures that heavily influential samples (high $I(W; Z_i)$) do not contribute too negatively to the bound, which is ensured to be non-vacuous. Moreover, contrarily to (1), a further application of Jensen's inequality and [6, Proposition 2] indicates that Corollary 1 compares positively to [4, Theorem 1], exhibiting the decaying factor of $1/\sqrt{n}$,

$$\left|\overline{\mathrm{gen}}(W, S)\right| \leq \frac{b - a}{n} \sum_{i=1}^{n} \mathbb{E}\left[\mathbb{W}(P_{W|Z_i}, P_W)\right] \leq (b - a)\Psi\left(\frac{I(W; S)}{n}\right) \leq \sqrt{\frac{(b - a)^2 I(W; S)}{2n}}. \tag{2}$$

It is also possible to obtain a random-subset version of [15, Theorem 2] using a similar strategy to [9, Propositions 2 and 4]. This kind of bounds, rather than looking at how knowing a *single sample* $Z_i$ modifies the hypothesis distribution, i.e., $\mathbb{W}(P_{W|Z_i}, P_W)$, look at how the knowledge of a set of samples $S_J$ alters the hypothesis distribution when all the other samples, $S_{J^c}$, used to obtain the hypothesis are known too, i.e., $W(P_{W|S}, P_{W|S_{J^c}})$.

**Theorem 2.** *Suppose that the loss function $\ell$ is $L$-Lipschitz for all $z \in \mathcal{Z}$ and that the hypothesis space $\mathcal{W}$ is Polish. Let $J$ be a uniformly random subset of $[n]$ such that $|J| = m$, and that is independent of $W$ and $S$. Let also $R$ be a random variable independent of $S$ and $J$. Then,*

$$\left|\overline{\mathrm{gen}}(W,S)\right| \leq L\mathbb{E}\big[\mathbb{W}(P_{W|S,R}, P_{W|S_{J^c},R})\big] \text{ and}$$

$$\left|\overline{\mathrm{gen}}(W,S)\right| \leq \frac{L}{m}\mathbb{E}\bigg[\sum_{i \in J}\mathbb{E}\big[\mathbb{W}(P_{W|S_{J^c}\cup Z_i,R}, P_{W|S_{J^c},R}) \mid J\big]\bigg].$$

In particular, when $m = 1$, the two equations from Theorem 2 reduce to [21, Lemma 3]

$$\left|\overline{\mathrm{gen}}(W,S)\right| \leq L\mathbb{E}\big[\mathbb{W}(P_{W|S,R}, P_{W|S^{-J},R})\big],$$

where $S^{-J} = S \setminus Z_J$, i.e., the whole dataset except sample $Z_J$. Moreover, if the loss is bounded and the geometry is ignored, Theorem 2 improves upon the tightest bounds in terms of the relative entropy of random subsets, cf. [7, Theorem 2.5].

**Corollary 2.** *In the conditions of Theorem 2, if the loss is bounded in $[a, b]$, then*

$$\left|\overline{\mathrm{gen}}(W,S)\right| \leq (b-a)\mathbb{E}\big[\mathrm{TV}(P_{W|S,R}, P_{W|S^{-J},R})\big] \leq \frac{b-a}{n}\sum_{j=1}^{n}\mathbb{E}\big[\Psi\big(D_{\mathrm{KL}}(P_{W|S,R} \parallel P_{W|S^{-j},R})\big)\big].$$

These data-dependent bounds characterize well the expected generalization error of the Langevin dynamics (LD) and stochastic gradient Langevin dynamics (SGLD) algorithms [7, Theorems 3.1 and 3.3], where $R$ is an artificial random variable used to encode some knowledge necessary to characterize the hypothesis distribution, such as the batch indices of SGLD. In particular, Corollary 2 improves upon [7, Theorem 2.5] tightening the elements of the expectation with respect to $J$ for which the divergence is large ($\gtrsim 1.6$).

It is possible to prove that Theorem 1 is tighter than [15, Theorem 1]. This results by studying the KR dual representation of the Wasserstein distance and noting that the conditional distribution $P_{W|Z_i}$ is a smoothed version of the forward channel, i.e., $P_{W|Z_i} = \mathbb{E}[P_{W|S}|Z_i]$. Comparisons with Theorem 2 are also possible using similar arguments and the triangle inequality. These results are informally summarized below and presented with more details and the proofs in Appendix D.1.

**Proposition 1.** *Consider the standard setting. Then, for all $j \subseteq [n]$ and all $i \in j$:*

$$\mathbb{E}\big[\mathbb{W}(P_{W|Z_i}, P_W)\big] \leq \mathbb{E}\big[\mathbb{W}(P_{W|S}, P_W)\big], \text{where } j = [n], \quad (\Longrightarrow \text{Theorem 1} \leq [15, \text{Theorem 1}])$$

$$\mathbb{E}\big[\mathbb{W}(P_{W|Z_i}, P_W)\big] \leq \mathbb{E}\big[\mathbb{W}(P_{W|S}, P_{W|S_{j^c}})\big], \text{and} \quad (\Longrightarrow \text{Theorem 1} \leq \text{Theorem 2})$$

$$\mathbb{E}\big[\mathbb{W}(P_{W|S}, P_{W|S_{j^c}})\big] \leq 2\mathbb{E}\big[\mathbb{W}(P_{W|S}, P_W)\big]. \quad (\Longrightarrow \text{Theorem 2} \leq 2\cdot[15, \text{Theorem 1}])$$

The following example showcases a situation where the presented bounds outperform the current known bounds based on the Wasserstein distance and the mutual information.

**Example 1** (Gaussian location model). *Consider the problem of estimating the mean $\mu$ of a $d$-dimensional Gaussian distribution with known covariance matrix $\sigma^2 I_d$. Further consider that there are $n$ samples $S = (Z_1, \ldots, Z_n)$ available, the loss is measured with the Euclidean distance $\ell(w, z) = \|w - z\|_2$, and the estimation is their empirical mean $W = \frac{1}{n}\sum_{i=1}^{n} Z_i$.*

In this example, the expected generalization error can be calculated exactly (see Appendix E):

$$\overline{\mathrm{gen}}(W,S) = \sqrt{\frac{2\sigma^2}{n}}\left(\sqrt{n+1} - \sqrt{n-1}\right)\frac{\Gamma\left(\frac{d+1}{2}\right)}{\Gamma\left(\frac{d}{2}\right)} \in \mathcal{O}\left(\frac{\sqrt{\sigma^2 d}}{n}\right).$$

As discussed in [6], the bound from [4] is not applicable in this setting since $I(W;S) \to \infty$ and since $\ell(w, Z)$ is not subgaussian given that $\mathrm{Var}[\ell(w, Z)] \to \infty$ as $\|w\|_2 \to \infty$. When $d = 1$, the loss $\ell(W, Z)$ is 1-subgaussian and the individual sample mutual information (ISMI) bound from [6] produces a bound in $\mathcal{O}\left(\sqrt{\sigma^2/n}\right)$, which decreases slower than the true generalization error, see Figure 1. This happens since the bound grows as the square root of $I(W; Z_i)$, which is in $\mathcal{O}(1/n)$.

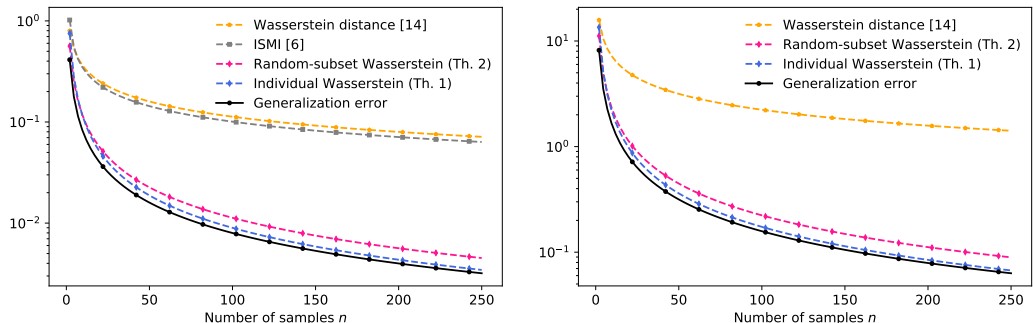

Figure 1: Expected generalization error and generalization error bounds for the Gaussian location model with $\mathcal{N}(\mu, 1)$ (left) and $\mathcal{N}(\mu, I_{250})$ (right). See Appendix E for the details.

In this scenario, the loss is 1-Lipschitz under $\rho(w, w') = \|w - w'\|_2$, and thus the bounds based on the Wasserstein distance are applicable. Applying the bound from [15] yields a bound in $\mathcal{O}\big(\sqrt{\sigma^2 d/n}\big)$, which decreases at the same sub-optimal rate as the ISMI bound. However, both the individual and random-subset Wasserstein distance bounds from Theorems 1 and 2 produce bounds in $\mathcal{O}\big(\sqrt{\sigma^2 d}/n\big)$, which decrease at the same rate as the true generalization error (see Figure 1).

### 3.1.1 Outline of the proofs

Similarly to [15, 16], the proofs of the theorems in this section are based on operating with $\overline{\text{gen}}(W, S)$ until an expression of the type $\mathbb{E}[f(X', Y) - f(X, Y)]$ is reached, where $X'$ is an independent copy of $X$ such that $P_{X',Y} = P_X \otimes P_Y$, and then applying the KR duality. For example, in Theorem 1 such an expression is achieved with $X = W$, $Y = Z_i$, and $f = \ell$. To arrive at these expressions, the proofs of Theorem 3 and 4 operate with $\overline{\text{gen}}(W, S)$ in different forms. More precisely,

(Th. 1) Since the samples $Z_i$ are independent and the expectation is a linear operator, the proof follows working with the quantity $\overline{\text{gen}}(W, S) = \frac{1}{n} \sum_{i=1}^{n} \mathbb{E}[\ell(W', Z_i) - \ell(W, Z_i)]$.

(Th. 2) Note that $\mathbb{E}[\mathscr{L}_{s_J}(w)] = \mathscr{L}_s(w)$, where $J$ is a uniformly random subset of $[n]$ of size $m$ and $s_J$ is the subset of $s$ indexed by $J$. This equality follows since there are $\binom{n}{m}$ subsets of size $m$ and each sample $z_i$ belongs to only $\binom{n-1}{m-1}$ of them. Hence,
$$\mathbb{E}[\mathscr{L}_{s_J}(w)] = \frac{1}{\binom{n}{m}} \sum_{j \in \mathcal{J}} \frac{1}{m} \sum_{i \in j} \ell(w, z_i) = \frac{1}{n} \sum_{i=1}^{n} \ell(w, z_i) = \mathscr{L}_s(w).$$
Then, the proof follows working with the quantity $\overline{\text{gen}}(W, \overline{S}) = \mathbb{E}[\mathscr{L}_{S_J}(W') - \mathscr{L}_{S_J}(W)]$.

### 3.2 Randomized-subsample setting

In the randomized-subsample setting the focus shifts from studying the impact of the samples on the hypothesis distribution to the impact of the samples' identities on the hypothesis distribution. For example, the analogous result to (1) is

$$\big|\overline{\text{gen}}(W, S)\big| \leq 2L\mathbb{E}\big[\mathbb{W}(P_{W|\tilde{S}, U}, P_{W|\tilde{S}})\big]. \tag{3}$$

Similarly to the standard setting, considering the discrete metric and applying Pinsker's and Jensen's inequalities leads to a less favorable bound than current bounds based on the mutual information [1, Theorem 5.1] since it does not explicitly decrease as $1/\sqrt{n}$. However, the bound still admits a tighter (see Appendix D.1) single-letter version.

**Theorem 3.** *Suppose that the loss function $\ell$ is $L$-Lipschitz for all $z \in \mathcal{Z}$ and that the hypothesis space $\mathcal{W}$ is Polish and let $\tilde{S}_i \triangleq (\tilde{Z}_i, \tilde{Z}_{i+n})$. Then,*

$$\big|\overline{\text{gen}}(W, S)\big| \leq \frac{2L}{n} \sum_{i=1}^{n} \mathbb{E}\big[\mathbb{W}(P_{W|\tilde{S}_i, U_i}, P_{W|\tilde{S}_i})\big].$$

As in the standard setting, when the loss function is bounded and the geometry of the space is ignored, this result improves upon current single-letter bounds based on the mutual information [9, 12] by

pulling the expectation with respect to the samples $P_{\tilde{S}_i}$ out of the square root. Here, the BH inequality is not considered since $D_{\mathrm{KL}}(P_{W|\tilde{S}_i,U_i} \| P_{W|\tilde{S}_i}) \leq \log(2)$; see Appendix F for the details.

**Corollary 3.** *Under the conditions of Theorem 3, if the loss $\ell$ is bounded in $[a, b]$, then*

$$\left|\overline{\mathrm{gen}}(W, S)\right| \leq \frac{2(b-a)}{n} \sum_{i=1}^{n} \mathbb{E}\left[\mathrm{TV}(P_{W|\tilde{S}_i,U_i}, P_{W|\tilde{S}_i})\right]$$

$$\leq \frac{b-a}{n} \sum_{i=1}^{n} \mathbb{E}\left[\sqrt{2D_{\mathrm{KL}}(P_{W|\tilde{S}_i,U_i} \| P_{W|\tilde{S}_i})}\right].$$

In this setting, Corollary 3 also decreases at a $1/\sqrt{n}$ rate and is tighter than [1, Theorem 5.1].

$$\left|\overline{\mathrm{gen}}(W, S)\right| \leq \frac{2(b-a)}{n} \sum_{i=1}^{n} \mathbb{E}\left[\mathbb{W}(P_{W|\tilde{Z}_i,\tilde{Z}_{i+n},U_i}, P_{W|\tilde{Z}_i,\tilde{Z}_{i+n}})\right] \leq \sqrt{\frac{2(b-a)^2 I(W;U|\tilde{S})}{n}}.$$

Finally, the randomized-subsample setting also accepts random-subset bounds. These bounds study how the knowledge of the *identities of a set of samples* that where used for training, $U_J$, alters the hypothesis distribution when all the other identities, $U_{J^c}$, and all the samples, $\tilde{S}$, are known.

**Theorem 4.** *Suppose that the loss function $\ell$ is $L$-Lipschitz for all $z \in \mathcal{Z}$ and that the hypothesis space $\mathcal{W}$ is Polish. Let $J$ be a uniformly random subset of $[n]$ such that $|J| = m$, and that is independent of $W$, $\tilde{S}$, and $U$ Let also $R$ be a random variable independent of $\tilde{S}, U$, and $J$. Then,*

$$\left|\overline{\mathrm{gen}}(W, S)\right| \leq 2L\mathbb{E}\left[\mathbb{W}(P_{W|\tilde{S},U,R}, P_{W|\tilde{S},U_{J^c},R})\right] \quad \text{and}$$

$$\left|\overline{\mathrm{gen}}(W, S)\right| \leq \frac{2L}{m}\mathbb{E}\left[\sum_{i \in J} \mathbb{E}\left[\mathbb{W}(P_{W|\tilde{S},U_{J^c}\cup U_i,R}, P_{W|\tilde{S},U_{J^c},R}) \mid J\right]\right].$$

Although these bounds are weaker than Theorem 3 (see Appendix D.1), their data-dependent nature may lead to more tractable and sharper bounds in practice. For example, when the discrete metric is considered, Theorem 4 recovers from below current random-subset bounds based on the relative entropy, which are used to obtain some of the tightest bounds for LD and SGLD [9, 10].

**Corollary 4.** *In the conditions of Theorem 4, for $m = 1$, if the loss is bounded in $[a, b]$, then*

$$\left|\overline{\mathrm{gen}}(W, S)\right| \leq 2(b-a)\mathbb{E}\left[\mathrm{TV}(P_{W|\tilde{S},U,R}, P_{W|\tilde{S},U^{-J},R})\right]$$

$$\leq \frac{b-a}{n} \sum_{j=1}^{n} \mathbb{E}\left[\sqrt{2D_{\mathrm{KL}}(P_{W|\tilde{S},U,R} \| P_{W|\tilde{S},U^{-j},R})}\right].$$

### 3.2.1 Outline of the proofs

The proofs of the results in this section are similar to those of the standard setting, hence their similar expressions. However, instead of operating with the expected generalization error in the form of $\mathbb{E}[\mathrm{gen}(W, S)]$ they operate with $\mathbb{E}[\widehat{\mathrm{gen}}(W, \tilde{S}, U)]$.

There are two issues that complicate the application of the KR duality as in the previous proofs. For instance, consider $\widehat{\mathrm{gen}}(W, \tilde{S}, U)$, then:

- Both $\mathscr{L}_{\bar{S}}(W)$ and $\mathscr{L}_S(W)$ depend on $P_U$. Hence, considering a copy $W'$ of $W$ such that $P_{W',\tilde{S},U} = P_{W,\tilde{S}} \otimes P_U$ does not help since $\mathbb{E}[\widehat{\mathrm{gen}}(W, \tilde{S}, U)] \neq \mathbb{E}[\mathscr{L}_{\bar{S}}(W') - \mathscr{L}_S(W)]$.
- Even if $\mathbb{E}[\widehat{\mathrm{gen}}(W, \tilde{S}, U)] = \mathbb{E}[\mathscr{L}_{\bar{S}}(W') - \mathscr{L}_S(W)]$ were true, for some fixed $\tilde{s}$ and $u$, the functions $\mathscr{L}_{\bar{s}}(w)$ and $\mathscr{L}_s(w)$ on $w$ are different, and thus the KR duality cannot be invoked.

Nonetheless, these two issues are resolved considering instead

$$\widehat{\mathrm{gen}}(W, \tilde{S}, U) = \mathscr{L}_{\bar{S}}(W) - \mathscr{L}_S(W) - \mathbb{E}[\mathscr{L}_{\bar{S}}(W') - \mathscr{L}_S(W')],$$

where $W'$ is an independent copy of $W$ such that $P_{W',\tilde{S},U} = P_{W,\tilde{S}} \otimes P_U$. Hence, the inequalities $\mathbb{E}[\mathscr{L}_{\bar{S}}(W') - \mathscr{L}_S(W')] = 0$ and $|x + y| \leq |x| + |y|$ lead to the upper bound

$$\left|\mathbb{E}[\widehat{\mathrm{gen}}(W, \tilde{S}, U)]\right| \leq \left|\mathbb{E}[\mathscr{L}_{\bar{S}}(W') - \mathscr{L}_{\bar{S}}(W)]\right| + \left|\mathbb{E}[\mathscr{L}_S(W') - \mathscr{L}_S(W)]\right|,$$

where the KR duality can be applied to each of the terms, albeit at the expense of an extra factor of 2.

### 3.3 Backward channel

In [17], the authors study the characterization of the expected generalization error in terms of the discrepancy between the data distribution $P_S$ and the backward channel distribution $P_{S|W}$ motivated by its connection to rate–distortion theory, see e.g., [19, Chapters 25–57] or [22, Chapter 10]. An approach formalizing this intuitive connection is given in Appendix G and different angles, based on chaining mutual information [13] and compression, are found in [11, Section 5] and [23].

More concretely, they proved that the generalization error is bounded from above by the discrepancy of these distributions, where the discrepancy is measured by the Wasserstein distance of order $p$ with the Minkowski distance of order $p$ as a metric, i.e., $\rho(x, y) = \|x - y\|_p$. Namely,

$$\left|\overline{\mathrm{gen}}(W, S)\right| \leq \frac{L}{n^{1/p}} \mathbb{E}[\mathbb{W}_{p,|\cdot|}^p(P_S, P_{S|W})]^{1/p}.$$

Similarly, the results from §3.1 and §3.2 can be replicated considering the backward channel instead of the forward channel, e.g., $P_{S|W}$ instead of $P_{W|S}$ in (1), $P_{Z_i|W}$ instead of $P_{W|Z_i}$ in Theorem 1. However, in this case, the loss $\ell$ would be required to be Lipschitz with respect to the samples space $\mathcal{Z}$ and not the hypothesis space $\mathcal{W}$, i.e., Lipschitz for all fixed $w \in \mathcal{W}$, thus exploiting the geometry of the samples' space and not the hypotheses' one.

As an example, noting that $\overline{\mathrm{gen}}(W, S) = \mathbb{E}[\mathscr{L}_{S'}(W) - \mathscr{L}_S(W)]$, where $S'$ is an independent copy of $S$ such that $P_{W,S'} = P_W \otimes P_S$ produces the bound

$$\left|\overline{\mathrm{gen}}(W, S)\right| \leq L\mathbb{E}[\mathbb{W}(P_S, P_{S|W})].$$

Compared to [17], these results (i) are valid for any metric $\rho$ as long as the loss $\ell$ is Lipschitz under $\rho$, and (ii) have single-letter and random-subset versions, and (iii) have variants in both the standard and randomized-subsample settings.

### 3.4 Other information measures

The bounds obtained in §3.1 and §3.2 may be manipulated to produce a variety of new bounds based on common information measures. For example, once the discrete metric is assumed and since the total variation is symmetric, applying Pinsker's inequality with the distributions in the opposite order to Corollaries 1, 2, 3, and 4 and further applying Jensen's inequality yields bounds based on the lautum information L [24]. For instance, a corollary of Theorem 3 is

$$\left|\overline{\mathrm{gen}}(W, S)\right| \leq \frac{b - a}{n} \sum_{i=1}^n \Psi\big(\mathrm{L}(W; Z_i)\big).$$

Similarly, several new bounds based on different *f-divergences* [19, Chapter 7] may be obtained employing the *joint range strategy* once the discrete metric is assumed. As an example, some corollaries of Theorem 1 based on the Hellinger distance H and the $\chi^2$-divergence (see Appendix H for a tighter and more general version of (6)) are

$$\left|\overline{\mathrm{gen}}(W, S)\right| \leq \frac{L}{2n} \sum_{i=1}^n \mathbb{E}\left[\mathrm{H}(P_{W|Z_i}, P_W)\sqrt{4 - \mathrm{H}^2(P_{W|Z_i}, P_W)}\right], \tag{4}$$

$$\left|\overline{\mathrm{gen}}(W, S)\right| \leq \frac{L}{\sqrt{2}n} \sum_{i=1}^n \mathbb{E}\left[\sqrt{\log\big(1 + \chi^2(P_{W|Z_i}, P_W)\big)}\right], \text{ and} \tag{5}$$

$$\left|\overline{\mathrm{gen}}(W, S)\right| \leq \frac{L}{2n} \sum_{i=1}^n \mathbb{E}\left[\sqrt{\chi^2(P_{W|Z_i}, P_W)}\right]. \tag{6}$$

### 3.5 Final remarks on the generality of the results

Due to Bobkov–Götze's theorem [14, Theorem 4.8], the relative entropy results still hold when the loss is both Lipschitz and subgaussian. Hence, the presented Wasserstein distance bounds are tighter than [4, 6, 7, 9, 10] in a more general setting. Moreover, the total variation results also hold for any metric with the added factor of $d_\rho(\mathcal{W})$ as per Remark 3. These results were omitted in the main text for clarity of exposition, but are included in Appendix B.

Therefore, only when the loss is not Lipschitz but is subgaussian or has a bounded cumulant function, or $\mathcal{W}$ is not Polish, the bounds from [6, 7, 10, 12] are preferred. As an example, some common loss functions such as the cross-entropy, the Hinge loss, the Huber loss, or any $L_p$ norm are Lipschitz [25, 26] under an appropriate metric $\rho$, see Appendix A for a discussion of the role of the metric and the space geometry in the presented bounds.

# 4 Discussion

This paper introduced several expected generalization error bounds based on the Wasserstein distance. In particular, these are full-dataset, single-letter, and random-subset bounds on both the standard and the randomized-subsample settings. When the Wasserstein distance ignores the geometry of the hypothesis space and the loss is bounded, the presented bounds are tighter and recover from below the current bounds based on the relative entropy and the mutual information [4, 6, 7, 9, 10], see also Appendix B for stronger, more general statements. Furthermore, the obtained total variation and relative-entropy bounds on the standard setting are ensured to be non-vacuous, i.e., smaller or equal than the trivial bound, thus resolving the issue of potentially vacuous relative-entropy and mutual-information bounds on the standard setting. Interestingly, the results for the randomized-subsample setting are tighter than their analogous in the standard setting only if their Wasserstein distance (or total variation) is twice as small.

Moreover, the techniques employed to obtain these bounds can also be used to obtain analogous bounds considering the backward channel and the samples' space geometry, aiming to facilitate connections between the generalization error characterization and rate–distortion theory, as suggested by Lopez and Jog [17]. Nonetheless, when the backward channel can be characterized, these bounds are interesting in their own right. Finally, the presented bounds may be used to generate a variety of new bounds in terms of, e.g., the lautum information or $f$-divergences like the total variation, the relative entropy, the Hellinger distance, or the $\chi^2$-divergence.

## 4.1 Limitations and future work

**PAC-Bayes bounds**   PAC-Bayes bounds ensure that $\mathbb{E}[\mathrm{gen}(W, S) \mid S] \geq \alpha(\beta^{-1})$ with probability no greater than $\beta \in (0, 1)$. Similarly, single-draw PAC-Bayes bounds ensure that $\mathrm{gen}(W, S) \geq \alpha(\beta^{-1})$ with probability no greater than $\beta \in (0, 1)$. These concentration bounds are of high probability when the dependency on $\beta^{-1}$ is logarithmic, i.e., $\log(1/\beta)$. See, [27, 2] for an overview.

The bounds from this work may be used to obtain single-draw PAC-Bayes bounds applying Markov's inequality [22, Problem 3.1] directly. For instance, employing it in Theorem 1 implies that

$$P_{W,S}\left(\mathrm{gen}(W, S) \geq \frac{L}{\beta n} \sum_{i=1}^{n} \mathbb{E}[\mathbb{W}(P_{W|Z_i}, P_W)]\right) \leq \beta,$$

for all $\beta \in (0, 1)$. However, this is not a high-probability bound since the dependency on $\beta^{-1}$ is linear. Hence, high-probability concentration bounds based on the Wasserstein distance and the total variation are a path of future research. As an example, [28–31] provide high-probability single-draw PAC-Bayes bounds based on, respectively, max-information, differential privacy, $\alpha$-mutual information, and uniform stability. Similarly, high-probability PAC-Bayes bounds based on the relative entropy and the hypothesis' space geometry are given in [8] and [32, 33], respectively.

**New bounds to specific algorithms**   The Wasserstein distance is difficult to characterize and/or estimate. Nonetheless, some of the bounds that can be obtained from it, e.g., mutual-information and relative-entropy bounds, have been used to obtain analytical bounds on specific algorithms, e.g., Langevin dynamics and stochastic gradient Langevin dynamics [6, 7, 9, 10]. Some of these results can be readily tightened with Corollaries 1, 2, and 4. Thence, deriving new analytical bounds for specific algorithms based on the presented results is also a topic for further research.

**Connections to stability and privacy measures**   A learning algorithm is said to be stable if a small change on the input dataset produces a small variation in the output hypothesis. There are various attempts at quantifying this notion such as uniform stability [34], where the variation in the output hypothesis is seen in terms of the loss, and differential privacy (DP) [28], where this variation is seen in terms of the hypothesis distribution. These notions are tied to the generalization capability of an algorithm, i.e., the less a hypothesis depends on the specifics of the data samples, the better it will generalize, and hence there are works obtaining generalization bounds based on stability, see e.g., [29, 31]. In particular, there are some works that, assuming some stability notion such as DP, bound from above the relative entropy and the mutual information appearing in some of the bounds that can be derived from the results presented in this work, hence also tying stability and generalization, c.f. [1, 35, 36]. Therefore, a future line of research is to investigate how different notions of stability can be combined with the measures of similarity between distributions employed in this work to characterize the generalization error.

## Funding

This work was funded in part by the Swedish research council under contract 2019-03606.

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
