# A    A short note on geometry and generalization error bounds

## A.1    Chaining- and Wasserstein-based bounds

Ever since it was shown that some infinitely-dimensional hypothesis spaces where PAC-learnable thanks to the VC dimension [2, Chapter 6], there has been an interest in studying the role of the space complexity and its geometry in determining the generalization of an algorithm, e.g. [37].

Some of the most interesting results come from the theory of random processes. Adapted to our notation, this theory considers the set of random variables (or random process) $\{\text{gen}(w, S)\}_{w \in \mathcal{W}}$. Then, this theory bounds the generalization error using the $\epsilon$-covering number of the hypothesis space $\mathcal{N}(\mathcal{W}, \rho, \epsilon)$ with some requirements on the smoothness of the hypothesis' space $\mathcal{W}$ under a metric $\rho$. Here, the geometry of the space captures its complexity via the $\epsilon$-covering number, which is defined as the cardinality of the minimum set $\mathcal{N}$ such that for all hypothesis $w$ in $\mathcal{W}$, there is an element of the set $x \in \mathcal{N}$ such that $\rho(x, w) \leq \epsilon$. The two main techniques from this line of work are:

- The Lipschitz maximal inequality [14, Lemma 5.7]. Here, the smoothness condition is to require the process to be Lipschitz; that is, that there is a random variable $C$ such that for all $w, w' \in \mathcal{W}$

$$|\text{gen}(w, S) - \text{gen}(w', S)| \leq C\rho(w, w').$$

  Then, with the additional condition that $\text{gen}(w, S)$ should be $\sigma$-subgaussian under $P_S$ for all $w \in \mathcal{W}$, this inequality bounds the generalization error as follows:

$$\mathbb{E}[\text{gen}(W, S)] \leq \inf_{\epsilon \in \mathbb{R}} \left\{ \epsilon \mathbb{E}[C] + \sqrt{2\sigma^2 \log \mathcal{N}(\mathcal{W}, \rho, \epsilon)} \right\}.$$

  Note how this technique creates a tension between the finesse $\epsilon$ of the covering and the smoothness of the process $\mathbb{E}[C]$.

- Dudley's chaining technique [14, Theorem 5.24]. Here, the smoothness condition is to require the process to be subgaussian; that is, for all $w, w' \in \mathcal{W}$ and all $\lambda \geq 0$,

$$\log \mathbb{E}\left[ e^{\lambda\left(\text{gen}(w,S) - \text{gen}(w',S)\right)} \right] \leq \frac{\lambda^2 \rho(w, w')^2}{2}$$

  and $\mathbb{E}[\text{gen}(w, S)] = 0$. Then, with the additional condition that the process is separable, this inequality bounds the generalization error as follows:

$$\mathbb{E}[\text{gen}(W, S)] \leq 6 \sum_{k \in \mathbb{Z}} 2^{-k} \sqrt{\log \mathcal{N}(\mathcal{W}, \rho, 2^{-k})}.$$

  Note that the subgaussian requirement is a relaxation of the Lipschitz requirement. As such, now the bound is expressed as a sum where each finer covering number of the space is weighted by the finesse $(2^{-k})$ of such a covering. Further refined bounds based on this technique can be found in [32, 33].

The first work that included the relationship between the hypothesis and the training samples to the analysis using random processes was [13]. There, the authors combined the chaining technique with [4, Lemma 1] to derive a formula which bounds the generalization error by a weighted average of the mutual information between the dataset and increasingly finer quantizations $W_k$ of the hypothesis. More precisely, they proved that

$$\mathbb{E}[\text{gen}(W, S)] \leq 3\sqrt{2} \sum_{k \in \mathbb{Z}} 2^{-k} \sqrt{I(W_k; S)}.$$

Therefore, in [13], the geometry of the hypothesis space $\mathcal{W}$ under the metric $\rho$ is expressed as the amount of information that the quantizations of the hypothesis under $\rho$ contain about the dataset $S$.

On the other hand, in [15] and this paper, a stronger smoothness requirement is considered. Namely that the loss function is $L$-Lipschitz; i.e., $|\ell(w, z) - \ell(w', z)| \leq L\rho(w, w')$ for all $w, w' \in \mathcal{W}$ and all $z \in \mathcal{Z}$. This is a stronger statement since the subgaussian assumption can be viewed as an "in-probability" version of the Lipschitz assumption. Nonetheless, this stronger assumption allows these bounds to bound the generalization error by the minimum cost to go from the distribution of the hypothesis after observing some samples to its marginal distribution, e.g., $\mathbb{W}(P_{W|Z_i}, P_W)$. Here the

metric $\rho$ is used to quantify how far away are the realizations of both probability distributions, on average, if they are coupled in the best way possible.

An interesting property of the presented bounds is that they have the flexibility to consider either the hypothesis or the sample space (backward channel). For instance, for [13, Example 1], the presented bounds using the backward channel are tighter than the bounds arising from the chaining technique.

In this example, the authors consider the canonical Gaussian process. The hypothesis space is $\mathcal{W} = \{w \in \mathbb{R}^2 : \|w\|_2 = 1\}$, the samples are $Z_i \sim \mathcal{N}(0, I_2)$, the loss function is $\ell(w, z) = -w^T z$, and the hypothesis is selected with the empirical risk minimization (ERM) algorithm, i.e., $w^\star = \arg\min_{w \in \mathcal{W}} \left\{ \frac{1}{n} \sum_{i=1}^n \ell(w, z_i) \right\}$. In this setting, by Cauchy–Schwarz we see that the loss $\ell(w, \cdot)$ is 1-Lipschitz for all $w \in \mathcal{W}$; i.e., $|-w^T z + w^T z'| \leq \|w\|_2 \|z - z'\|_2 \leq \|z - z'\|_2$ for all $z, z' \in \mathcal{Z}$. Therefore, the backward channel equivalent of Theorem 1 holds. Moreover, the function is 1-subgaussian under $P_Z$ for all $w \in \mathcal{W}$. Therefore, as shown in Appendix B.1 the presented bound is tighter than [23], which is shown to be tighter than [13] in this setting (see [23, Section IV-B]).

### A.2 Choice of the metric

The presented bounds in Sections 3.1, 3.2, and 3.3 are valid for any metric $\rho$ under which the loss is Lipschitz. The Lipschitz property is a property of the hypothesis space $\mathcal{W}$ (forward channel bounds) or the sample space $\mathcal{Z}$ (backward channel bounds) and *not* of the algorithm. That is, if two different algorithms operate on the same sample space and produce the same hypothesis space, then they can be characterized with the same metric.

The choice of the metric can be decisive for a tight analysis of the presented bounds, and there are times where a loss function can be Lipschitz under several metrics. For example, a bounded loss function represented as a norm is Lipschitz with respect to that norm and the discrete metric. Nonetheless, in many situations the metric of choice becomes apparent based on the loss function. For example, if we consider the forward channel bounds and samples of the type $z = (x, y)$ and the following two common supervised tasks:

- Regression. If a norm is used as the loss function $\ell(w, z) = \|w - y\|$, then such a norm is also a good choice for a metric since by the reverse triangle inequality the loss is 1-Lipschitz under that metric: $\big| \|w - y\| - \|w' - y\| \big| \leq \|w - w'\|$ for all $w, w' \in \mathcal{W}$.

- Classification. If the 0-1 loss is used as the loss function $\ell(w, z) = \mathrm{Ind}(w \neq y)$, then the discrete metric is a good choice since the loss is also 1-Lipschitz under this metric: $\big| \mathrm{Ind}(w \neq y) - \mathrm{Ind}(w' \neq y) \big| \leq \mathrm{Ind}(w \neq w')$.

Similarly, for the backward channel bounds, it is known that the logistic loss, the softmax loss,[2] the Hinge loss, and many distance-based losses like norms, the Huber, $\epsilon$-insensitive, and pinball losses, are Lipschitz under the $L_1$ norm metric $\rho(z, z') = |z - z'|$ [25, 26].

## B   Generality of the results

The main text only presents total variation and relative entropy bounds for bounded losses. This is to clarify the (geometrical) relationship between the bounds based on the Wasserstein distance and those based on the relative entropy. That is, that the former recover the latter when the geometry of the hypothesis space is ignored.

Nonetheless, these bounds hold more generally for any metric under certain mild modifications. More concretely, the relative entropy bounds hold when the loss is also subgaussian[3]. Furthermore, at the cost of an extra factor of $d_\rho(\mathcal{W})$, the two kinds of bound still hold.

### B.1 Extension of the relative entropy bounds to subgaussian losses

Consider a Polish space $(\mathcal{X}, \rho)$ and a probability distribution $P$ on $\mathcal{X}$ with a finite first moment. Then, the Bobkov–Götze's theorem [14, Theorem 4.8] says that the following statements are equivalent:

---

[2]This result can be derived from [26, Proposition 3] and the $L_1 - L_2$ inequality.

[3]A random variable $X$ is said to be $\sigma$-subgaussian if $\log \mathbb{E}[\exp \lambda(X - \mathbb{E}[X])] \leq \frac{\lambda^2 \sigma^2}{2}$ for all $\lambda \in \mathbb{R}$. Also, a function $f : \mathcal{X} \to \mathbb{R}$ is said to be $\sigma$-subgaussian under $P$ if $f(X)$ is $\sigma$-subgaussian and $X \sim P$.

- $f$ is $\sigma$-subgaussian under $P$ *for every* 1-Lipschitz function $f : \mathcal{X} \to \mathbb{R}$; and,

- $\mathbb{W}(Q, P) \le \sqrt{2\sigma^2 D_{\mathrm{KL}}(Q \parallel P)}$ for all $Q$ on $\mathcal{X}$.

Therefore, the results from Corollaries 1, 2, 3, and 4 are valid in a more general setting, namely when the loss function $\ell$ is both $L$-Lipschitz and $\sigma$-subgaussian for all $z \in \mathcal{Z}$. To realize this, note that if $X \sim P$ is $L$-Lipschitz and $\sigma$-subgaussian, then $X/L$ is 1-Lipschitz and $(\sigma/L)$-subgaussian. Hence, if $|X - \mathbb{E}[X]| \le L\mathbb{W}(Q, P)$, then it follows that $|X - \mathbb{E}[X]| \le L\sqrt{2(\sigma/L)^2 D_{\mathrm{KL}}(Q \parallel P)} = \sqrt{2\sigma^2 D_{\mathrm{KL}}(Q \parallel P)}$.

As an example, assume that the loss function $\ell$ is $L$-Lipschitz and $\sigma$-subgaussian under $P_W$ for all $z \in \mathcal{Z}$ and that $\mathcal{W}$ is Polish. Then,

$$\left|\overline{\mathrm{gen}}(W, S)\right| \le \frac{L}{n} \sum_{i=1}^{n} \mathbb{E}\left[\mathbb{W}(P_{W|Z_i}, P_W)\right] \le \frac{1}{n} \sum_{i=1}^{n} \sqrt{2\sigma^2 I(W; Z_i)} \le \sqrt{\frac{2\sigma^2 I(W; S)}{n}}$$

is a corollary of Theorem 1 due to Bobkov–Götze's theorem. As when the loss is bounded, this equation shows that Theorem 1 is tighter than [6, Proposition 2] and [4, Theorem 1] and exhibits a decaying factor of $1/\sqrt{n}$. Moreover, this result encompasses the case where the loss is bounded in $[a, b]$ since if a random variable is bounded in $[a, b]$ it is $(b - a)/2$-subgaussian.

Note, however, that in this case the subgaussianity constant is different than the constant from, e.g. [4], where the loss $\ell$ was supposed to be $\nu$-subgaussian under $P_Z$ for all $w \in \mathcal{W}$. Considering the bounds based on the backward channel (§3.3), these constants are exactly the same, since assuming that the loss function $\ell$ is $L$-Lipschitz and $\nu$-subgaussian under $P_Z$ for all $w \in \mathcal{W}$ and that $\mathcal{Z}$ is Polish, means that

$$\left|\overline{\mathrm{gen}}(W, S)\right| \le \frac{L}{n} \sum_{i=1}^{n} \mathbb{E}\left[\mathbb{W}(P_{Z_i|W}, P_{Z_i})\right] \le \frac{1}{n} \sum_{i=1}^{n} \sqrt{2\nu^2 I(W; Z_i)} \le \sqrt{\frac{2\nu^2 I(W; S)}{n}},$$

which is always tighter than [4, 6]. As mentioned above, when the loss is bounded the subgaussianity constant is the same under any distribution.

## B.2 Extension to the total variation bounds for any metric

As per Remark 3, the results based on the total variation also hold for any metric with the extra factor $d_\rho(\mathcal{W})$, where $d_\rho(\mathcal{W})$ is the diameter of the hypothesis space $\mathcal{W}$ under the metric $\rho$. For instance, Corollary 1 results in

$$\left|\overline{\mathrm{gen}}(W, S)\right| \le \frac{L d_\rho(\mathcal{W})}{n} \sum_{i=1}^{n} \mathbb{E}\left[\mathrm{TV}(P_{W|Z_i}, P_W)\right] \le \frac{L d_\rho(\mathcal{W})}{n} \sum_{i=1}^{n} \mathbb{E}\left[\Psi\left(D_{\mathrm{KL}}(P_{W|Z_i} \parallel P_W)\right)\right].$$

Note that, since the results based on the total variation hold for any metric, all the derived results based on different information measures from §3.4 hold too.

The extra term can be arbitrarily large as, for example, when the hypothesis is the weights of a neural network and metric $\rho$ is the $\ell_2$ norm $\|\cdot\|_2$. Nonetheless, this term can still be small and relevant for practical settings. For instance, consider again that the hypothesis is the weights of a neural network. However, consider now that the metric $\rho$ is the infinity norm $\|\cdot\|_\infty$ and that each weight is enforced to be smaller than some small constant $C$. Then, the diameter of the space $d_{\|\cdot\|_\infty}(\mathcal{W})$ is (at most) equal to $C$.

# C Proofs of the theorems from Section 3

## C.1 Proof of Theorem 1

Note that $\mathbb{E}[\mathscr{L}_{P_Z}(W)] = \int_{\mathcal{W}\times\mathcal{Z}} \ell(w,z) d(P_W \otimes P_Z)(w,z)$. If $W'$ is an independent copy of $W$ such that $P_{W',Z_i} = P_W \otimes P_Z$ for all $i \in [n]$, then

$$
\begin{aligned}
\left|\overline{\mathrm{gen}}(W,S)\right| &= \left|\frac{1}{n}\sum_{i=1}^{n}\mathbb{E}\big[\ell(W',Z_i) - \ell(W,Z_i)\big]\right| \\
&= \left|\frac{1}{n}\sum_{i=1}^{n}\mathbb{E}\Big[\mathbb{E}\big[\ell(W',Z_i) - \ell(W,Z_i) \mid Z_i\big]\Big]\right| \\
&\leq \frac{L}{n}\sum_{i=1}^{n}\mathbb{E}\big[\mathbb{W}(P_{W|Z_i}, P_W)\big],
\end{aligned}
$$

where the last inequality stems from the KR duality and the Lipschitzness of $\ell$ for all $z \in \mathcal{Z}$. The absolute value is removed since $\rho$ is a metric.

## C.2 Proof of Theorem 2

Consider the quantity

$$
\mathrm{gen}_j(w,s_j) \triangleq \mathscr{L}_{P_Z}(w) - \mathscr{L}_{s_j}(w),
$$

where $\mathscr{L}_{s_j} = \frac{1}{m}\sum_{i\in j}\ell(w,z_i)$. Then, note that $\mathbb{E}[\mathrm{gen}(W,S)] = \mathbb{E}[\mathrm{gen}_J(W,S_J)]$ if $\mathscr{L}_s(w) = \mathbb{E}[\mathscr{L}_{s_J}(w)]$. This last equality follows since $J$ is uniformly distributed, there are $\binom{n}{m}$ possible subsets of size $m$ in $[n]$, and each sample $z_i$ belongs to $\binom{n-1}{m-1}$ of those subsets; hence

$$
\mathbb{E}[\mathscr{L}_{s_J}(w)] = \frac{1}{\binom{n}{m}}\sum_{j\in\mathcal{J}}\frac{1}{m}\sum_{i\in j}\ell(w,z_i) = \frac{1}{n}\sum_{i=1}^{n}\ell(w,z_i) = \mathscr{L}_s(w). \tag{7}
$$

Subsequently, the two bounds from the theorem are obtained after bounding the innermost expectation on the right hand side of the following equation:

$$
\left|\overline{\mathrm{gen}}(W,S)\right| = \left|\mathbb{E}[\mathrm{gen}_J(W,S_J)]\right| \leq \mathbb{E}\big[\left|\mathbb{E}[\mathrm{gen}_J(W,S_J) \mid J, S_{J^c}, R]\right|\big],
$$

where the last step follows from Jensen's inequality.

(a) Consider, until stated otherwise, that all random objects and expectations are conditioned to a fixed $j$, $s_{j^c}$, and $r$. Then note that $\mathbb{E}[\mathscr{L}_{P_Z}(W)] = \mathbb{E}[\mathscr{L}_{S_j}(W')]$, where $W'$ is an independent copy of $W$ such that $P_{W',S_j|s_{j^c},r} = P_{W|s_{j^c},r} \otimes P_{S_j}$. Therefore

$$
\left|\mathbb{E}[\mathrm{gen}_j(W,S_j)]\right| \leq \left|\mathbb{E}[\mathscr{L}_{S_j}(W') - \mathscr{L}_{S_j}(W)]\right| \leq L\mathbb{E}[\mathbb{W}(P_{W|S_j,s_{j^c},r}, P_{W|s_{j^c},r})],
$$

where the last inequality stems from the KR duality. Finally, taking the expectation with respect to $P_{J,S_{J^c},R}$ in both sides of the equation completes the proof.

(b) Consider again, until stated otherwise, that all random objects and expectations are conditioned to a fixed $j$, $s_{j^c}$, and $r$. Then, similarly to the proof of Theorem 1

$$
\begin{aligned}
\left|\mathbb{E}[\mathrm{gen}_j(W,S_j)]\right| &\leq \left|\frac{1}{m}\sum_{i\in j}\mathbb{E}[\ell(W',Z_i) - \ell(W,Z_i)]\right| \\
&\leq \frac{L}{m}\sum_{i\in j}\mathbb{E}[\mathbb{W}(P_{W|Z_i,s_{j^c},r}, P_{W|s_{j^c},r})],
\end{aligned}
$$

where the last inequality stems from the KR duality and $W'$ is an independent copy of $W$ such that $P_{W',Z_i|s_{j^c},r} = P_{W'|s_{j^c},r} \otimes P_{Z_i}$ for all $i \in j$. Finally, taking the expectation with respect to $P_{J,S_{J^c},R}$ in both sides of the equation completes the proof.

## C.3 Proof of Equation 3

Consider an independent copy $W'$ of $W$ such that $P_{W',\tilde{S},U} = P_{W',\tilde{S}} \otimes P_U$. Then, $\mathbb{E}[\mathscr{L}_S(W')] = \mathbb{E}[\mathscr{L}_{\bar{S}}(W')]$, and therefore

$$\widehat{\mathrm{gen}}(w,\tilde{s},u) = \mathscr{L}_{\bar{s}}(w) - \mathscr{L}_s(w) - \mathbb{E}[\mathscr{L}_{\bar{S}}(W') - \mathscr{L}_S(W')].$$

Then, re-arranging the expectation of the above expression and using the fact that $|x + y| \leq |x| + |y|$ results in

$$\begin{aligned}
\left|\mathbb{E}[\widehat{\mathrm{gen}}(W,\tilde{S},U)]\right| &\leq \left|\mathbb{E}[\mathscr{L}_{\bar{S}}(W') - \mathscr{L}_{\bar{S}}(W)]\right| + \left|\mathbb{E}[\mathscr{L}_S(W') - \mathscr{L}_S(W)]\right| \\
&= \left|\mathbb{E}\left[\mathbb{E}[\mathscr{L}_{\bar{S}}(W') - \mathscr{L}_{\bar{S}}(W) \mid \tilde{S},U]\right]\right| + \left|\mathbb{E}\left[\mathbb{E}[\mathscr{L}_S(W') - \mathscr{L}_S(W) \mid \tilde{S},U]\right]\right| \\
&\leq 2L\mathbb{E}[\mathbb{W}(P_{W|\tilde{S},U}, P_{W|\tilde{S}})],
\end{aligned}$$

where the last step stems from the KR duality. Finally, noting that $\mathbb{E}[\widehat{\mathrm{gen}}(W,\tilde{S},U)] = \overline{\mathrm{gen}}(W,S)$ completes the proof.

## C.4 Proof of Theorem 3

Similarly to the proof of (1), consider an independent copy $W'$ of $W$ such that $P_{W',\tilde{S}_i,U_i} = P_{W',\tilde{S}_i} \otimes P_{U_i}$. Then $\mathbb{E}[\ell(W',\bar{Z}_i)] = \mathbb{E}[\ell(W',Z_i)]$, where $Z_i = \tilde{Z}_{i+U_in}$, $\bar{Z}_i = \tilde{Z}_{i+(1-U_i)n}$, and $\tilde{S}_i = (Z_i,\bar{Z}_i)$. Therefore,

$$\widehat{\mathrm{gen}}(w,\tilde{s},u) = \frac{1}{n}\sum_{i=1}^{n}\left(\ell(w,\bar{z}_i) - \ell(w,z_i) - \mathbb{E}[\ell(W',\bar{Z}_i) - \ell(W',Z_i)]\right).$$

Then, re-arranging the expectation of the above expression and using the fact that $\left|\sum_{i=1}^n x_i\right| \leq \sum_{i=1}^n |x_i|$ results in

$$\begin{aligned}
\left|\widehat{\mathrm{gen}}(W,\tilde{S},U)\right| &\leq \frac{1}{n}\sum_{i=1}^{n}\left(\left|\mathbb{E}[\ell(W',\bar{Z}_i) - \ell(W,\bar{Z}_i)]\right| + \left|\mathbb{E}[\ell(W',Z_i) - \ell(W,Z_i)]\right|\right) \\
&= \frac{1}{n}\sum_{i=1}^{n}\left(\left|\mathbb{E}\left[\mathbb{E}[\ell(W',\bar{Z}_i) - \ell(W,\bar{Z}_i) \mid \tilde{S}_i,U_i]\right]\right| + \left|\mathbb{E}\left[\mathbb{E}[\ell(W',Z_i) - \ell(W,Z_i) \mid \tilde{S}_i,U_i]\right]\right|\right) \\
&\leq \frac{2L}{n}\sum_{i=1}^{n}\mathbb{E}\left[\mathbb{W}(P_{W|\tilde{S}_i,U_i}, P_{W|\tilde{S}_i})\right],
\end{aligned}$$

where the last step stems from the KR duality. Finally, noting that $\mathbb{E}[\widehat{\mathrm{gen}}(W,\tilde{S},U)] = \overline{\mathrm{gen}}(W,S)$ completes the proof.

## C.5 Proof of Theorem 4

Similarly to the proof of Theorem 2, consider the quantity

$$\widehat{\mathrm{gen}}_j(w,\tilde{s}_j,u_j) \triangleq \mathscr{L}_{\bar{s}_j}(w) - \mathscr{L}_{s_j}(w),$$

where $\tilde{s}_j = (s_j,\bar{s}_j)$ and $s_j$ and $\bar{s}_j$ are the subsets of $s$ and $\bar{s}$, respectively, indexed by $j$. Then, note that $\mathbb{E}[\widehat{\mathrm{gen}}(W,\tilde{S},U)] = \mathbb{E}[\widehat{\mathrm{gen}}_J(W,\tilde{S}_J,U_J)]$ since, as shown in (7), the equalities $\mathbb{E}[\mathscr{L}_{s_j}(w)] = \mathscr{L}_s(w)$ and $\mathbb{E}[\mathscr{L}_{\bar{s}_j}(w)] = \mathscr{L}_{\bar{s}}(w)$ hold.

Then, the two bounds from the theorem are obtained after bounding the innermost expectation on the right hand side of the following equation:

$$\left|\mathbb{E}[\widehat{\mathrm{gen}}(W,\tilde{S},U)]\right| = \left|\mathbb{E}[\widehat{\mathrm{gen}}_J(W,\tilde{S}_J,U_J)]\right| \leq \mathbb{E}\left[\left|\mathbb{E}[\widehat{\mathrm{gen}}_J(W,\tilde{S}_J,U_J) \mid J,\tilde{S},U_{J^c},R]\right|\right],$$

where the last step follows from Jensen's inequality.

(a) Consider, until stated otherwise, that all random objects and expectations are conditioned to a fixed $j$, $\tilde{s}$, $u_{j^c}$, and $r$. Then note that $\mathbb{E}[\mathscr{L}_{\bar{s}_j}(W')] = \mathbb{E}[\mathscr{L}_{s_j}(W')]$, where $W'$ is an independent copy of $W$ such that $P_{W',U_j|\tilde{s},u_{j^c},r} = P_{W'|\tilde{s},u_{j^c},r} \otimes P_{U_j}$. Therefore,

$$\widehat{\text{gen}}_j(w, \tilde{s}_j, u_j) = \mathscr{L}_{\bar{s}_j}(w) - \mathscr{L}_{s_j}(w) - \mathbb{E}[\mathscr{L}_{\bar{s}_j}(W') - \mathscr{L}_{s_j}(W')].$$

Then, re-arranging the expectation of the above expression and using the fact that $|x + y| \leq |x| + |y|$ results in

$$\begin{aligned}
\left|\widehat{\text{gen}}_j(W, \tilde{s}_j, U_j)\right| &\leq \left|\mathbb{E}[\mathscr{L}_{\bar{s}_j}(W') - \mathscr{L}_{\bar{s}_j}(W)]\right| + \left|\mathbb{E}[\mathscr{L}_{s_j}(W') - \mathscr{L}_{s_j}(W)]\right| \\
&= \left|\mathbb{E}\big[\mathbb{E}[\mathscr{L}_{\bar{s}_j}(W') - \mathscr{L}_{\bar{s}_j}(W) \mid U_j]\big]\right| + \left|\mathbb{E}\big[\mathbb{E}[\mathscr{L}_{s_j}(W') - \mathscr{L}_{s_j}(W) \mid U_j]\big]\right| \\
&\leq 2L\mathbb{E}[\mathbb{W}(P_{W|U_j,\tilde{s},u_{j^c},r}, P_{W|\tilde{s},u_{j^c},r})],
\end{aligned}$$

where the last inequality stems from the KR duality. Finally, taking the expectation with respect to $P_{J,\tilde{S},U_{J^c},R}$ in both sides of the equation completes the proof.

(b) Consider again, until stated otherwise, that all random objects and expectations are conditioned to a a fixed $j$, $\tilde{s}$, $u_{j^c}$, and $r$. Then, similarly to the proof of Theorem 3

$$\begin{aligned}
\left|\widehat{\text{gen}}_j(W, \tilde{s}, U_j)\right| &\leq \frac{1}{m} \sum_{i \in j} \left( \left|\mathbb{E}[\ell(W', \bar{Z}_i) - \ell(W, \bar{Z}_i)]\right| + \left|\mathbb{E}[\ell(W', Z_i) - \ell(W, Z_i)]\right| \right) \\
&= \frac{1}{m} \sum_{i \in j} \left( \left|\mathbb{E}\big[\mathbb{E}[\ell(W', \bar{Z}_i) - \ell(W, \bar{Z}_i) \mid U_i]\big]\right| + \left|\mathbb{E}\big[\mathbb{E}[\ell(W', Z_i) - \ell(W, Z_i) \mid U_i]\big]\right| \right) \\
&\leq \frac{2L}{m} \sum_{i \in j} \mathbb{E}\big[\mathbb{W}(P_{W|U_i,\tilde{s},u_{j^c},r}, P_{W|\tilde{s},u_{j^c},r})\big],
\end{aligned}$$

where the last inequality stems from the KR duality. Finally, taking the expectation with respect to $P_{J,\tilde{S},U_{J^c},R}$ in both sides of the equation completes the proof.

# D Comparison of the bounds

In this section of the appendix, the full-dataset, single-letter, and random-subset bounds are compared. First, remember that the bounds based on the Wasserstein distance are tighter than the respective ones based on the relative entropy, and thus, those based on the mutual information under the conditions specified in Appendix B. The comparison in terms of the Wasserstein distance and in terms of the mutual information are found in §D.1 and in D.2, respectively.

When the bounds are compared in terms of the mutual information and in terms of the Wasserstein distance, the individual-sample bounds are shown to be tighter than the full-dataset bounds. Therefore, the idea that the individual forward channels $P_{W|Z_i}$, which are smoothed versions of the full-dataset forward channel $P_{W|S}$, are closer to the hypothesis marginal distribution $P_W$ is backed by the theory in both cases.

Then, both cases also agree in that individual-sample bounds are tighter than random-subset bounds. Even though the proof for the Wasserstein distance based bounds also follows from a smoothing argument, a better insight is gained through the proof for the mutual information. These are based on the fact that $I(W; Z_i) \leq I(W; Z_i|S_{\mathcal{A}})$, where $\mathcal{A}$ is a subset of $[n]$ where $i$ is not included. This inequality holds since the knowledge of the samples $S_{\mathcal{A}}$ provides information about $Z_i$ through $W$. More precisely,

$$\begin{aligned}
I(W; Z_i) &\overset{(a)}{\leq} I(W; Z_i) + \overbrace{I(S_{\mathcal{A}}; Z_i|W)}^{\text{extra information}} \\
&\overset{(b)}{=} I(W, S_{\mathcal{A}}; Z_i) \\
&\overset{(c)}{=} I(W; Z_i|S_{\mathcal{A}}) + I(Z_i; S_{\mathcal{A}}) \\
&\overset{(d)}{=} I(W; Z_i|S_{\mathcal{A}}),
\end{aligned}$$

where $(a)$ is due to the non-negativity of the mutual information, $(b)$ and $(c)$ follow from the chain rule, and $(d)$ stems from the fact that $Z_i$ and $S_{\mathcal{A}}$ are independent.

Finally, the comparison of the random-subset and full-dataset bounds behaves differently when the Wasserstein distance or the mutual information is employed. The analysis using the Wasserstein distance suggests that the random-subset bounds are sharper than the full-dataset bounds with an extra factor of two, namely

$$\mathbb{E}[\mathbb{W}(P_{W|S}, P_{W|S_{J^c}})] \leq 2\mathbb{E}[\mathbb{W}(P_{W|S}, P_W)].$$

On the other hand, the analysis using the mutual information indicates that the random-subset bounds are looser than the full dataset bounds.

This discrepancy might help understand the reason why, in practice, the random-subset bounds from [7, 10, 9] with the expectation of the square root of the relative entropy result in tighter characterizations of the generalization error than the full-dataset bounds from [4, 1] using the mutual information. To be precise, this suggests that the loss of performance of a further application of the Jensen's inequality to incorporate the expectations inside the square root of the bounds from [7, 10, 9] is big. In other words, that $D_{\mathrm{KL}}(P_{W|S} \parallel P_{W|S_J})$ has high variance in practical settings.

A summary of these comparisons is shown in Figure 2. Note that the mutual information bounds are written after Jensen's inequality is applied in order to allow comparisons between them. However, the relationships between the Wasserstein and mutual information based bounds still hold when the expectations of the relative entropy are outside of the square root.

### D.1 Comparison of the Wasserstein distance based bounds

#### D.1.1 Standard setting

**Proposition 2.** *Consider the standard setting. Then,*

$$\frac{1}{n} \sum_{i=1}^{n} \mathbb{E}\big[\mathbb{W}(P_{W|Z_i}, P_W)\big] \leq \mathbb{E}\big[\mathbb{W}(P_{W|S}, P_W)\big].$$

*Proof.* The proposition follows by noting that, for all $i \in [n]$,

$$\mathbb{E}\big[\mathbb{W}(P_{W|Z_i}, P_W)\big] \leq \mathbb{E}\big[\mathbb{W}(P_{W|S}, P_W)\big], \tag{8}$$

which is a stronger statement than the original. More precisely, it can be shown that, for all $i \in [n]$,

$$\mathbb{E}\left[\sup_{f \in 1\text{-Lip}(\rho)} \big\{ \mathbb{E}[f(W) \mid Z_i] - \mathbb{E}[f(W)] \big\}\right] \leq \mathbb{E}\left[\sup_{f \in 1\text{-Lip}(\rho)} \big\{ \mathbb{E}[f(W) \mid S] - \mathbb{E}[f(W)] \big\}\right],$$

which is equivalent to (8) due to the KR duality.

After writing $\mathbb{E}\big[\sup_{f \in 1\text{-Lip}(\rho)} \big\{ \mathbb{E}[f(W) \mid S] - \mathbb{E}[f(W)] \big\}\big]$ in integral form, the result is shown as follows:

$$\int_{\mathcal{Z}^n} \sup_{f \in 1\text{-Lip}(\rho)} \left\{ \int_{\mathcal{W}} f(w) dP_{W|s}(w) - \int_{\mathcal{W}} f(w) dP_W(w) \right\} dP_S(s)$$

$$\overset{(a)}{\geq} \int_{\mathcal{Z}} \sup_{f \in 1\text{-Lip}(\rho)} \left\{ \int_{\mathcal{Z}^{n-1}} \left( \int_{\mathcal{W}} f(w) dP_{W|s}(w) - \int_{\mathcal{W}} f(w) dP_W(w) \right) P_Z^{\otimes n-1}(s^{-i}) \right\} dP_Z(z_i)$$

$$\overset{(b)}{=} \int_{\mathcal{Z}} \sup_{f \in 1\text{-Lip}(\rho)} \left\{ \int_{\mathcal{W}} f(w) dP_{W|z_i}(w) - \int_{\mathcal{W}} f(w) dP_W(w) \right\} dP_Z(z_i),$$

where $s^{-i} = (z_1, \ldots, z_{i-1}, z_{i+1}, \ldots, z_n)$, $(a)$ is due to the fact that $\sup_g \mathbb{E}[g(X)] \leq \mathbb{E}[\sup_g g(X)]$, and $(b)$ follows from Fubini–Tonelli's theorem and the fact that $P_{W|Z_i} = \mathbb{E}[P_{W|S} \mid Z_i]$.

Finally, noting that $(b)$ is the integral form of $\mathbb{E}\big[\sup_{f \in 1\text{-Lip}(\rho)} \big\{ \mathbb{E}[f(W) \mid Z_i] - \mathbb{E}[f(W)] \big\}\big]$ concludes the proof. $\qquad \square$

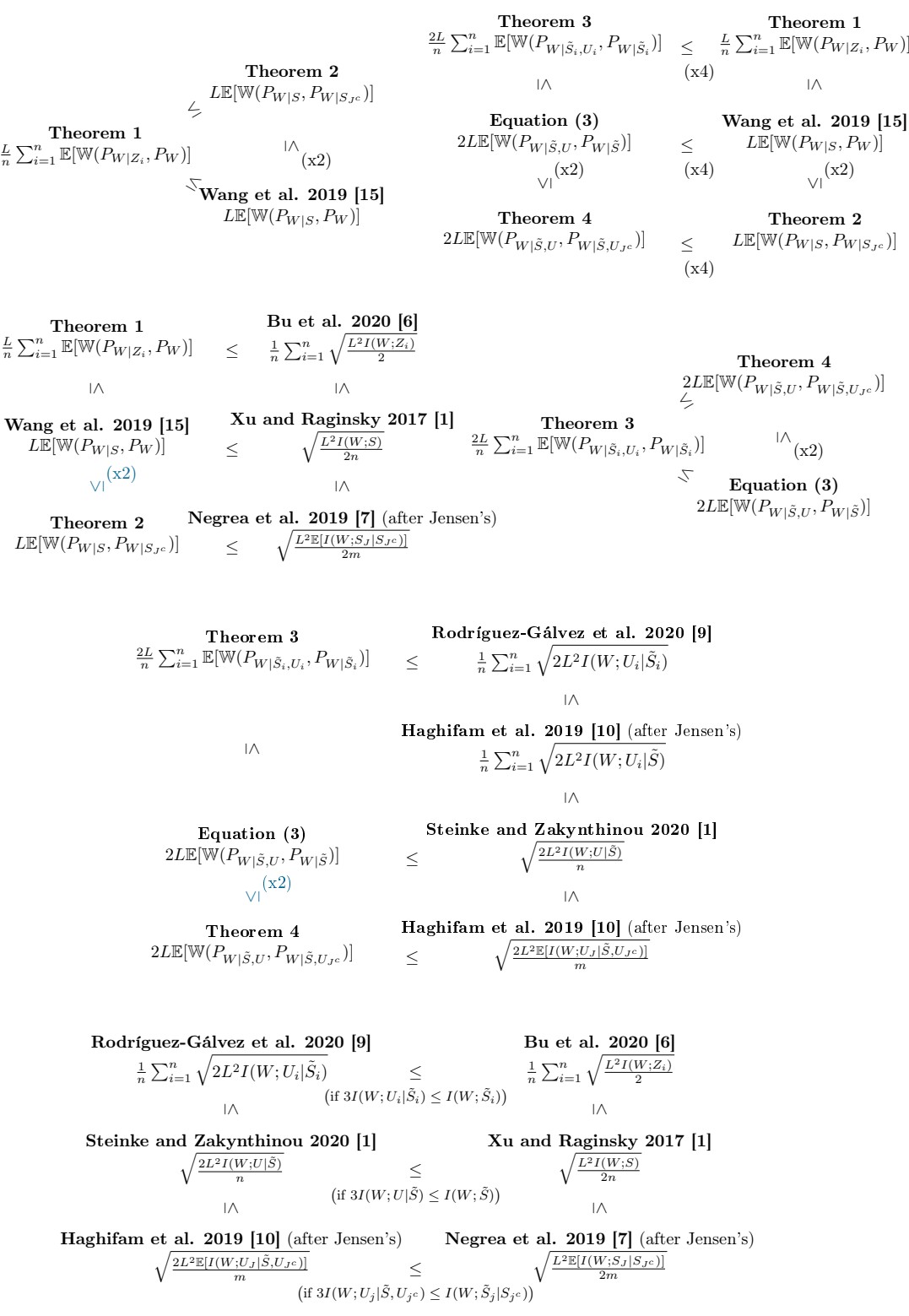

Figure 2: Summary of the comparison between the current and presented bounds based on the Wasserstein distance and the mutual information.

**Proposition 3.** *Consider the standard setting. Consider also a uniformly random subset of indices $J \subseteq [n]$ of size $m$. Then,*

$$\frac{1}{n} \sum_{i=1}^{n} \mathbb{E}\big[\mathbb{W}(P_{W|Z_i}, P_W)\big] \leq \mathbb{E}\big[\mathbb{W}(P_{W|S}, P_{W|S_{J^c}})\big].$$

*Proof.* The proof of this proposition follows closely that of Proposition 2. First note that the statement may be written as

$$\frac{1}{n} \sum_{i=1}^{n} \mathbb{E}\big[\mathbb{W}(P_{W|Z_i}, P_W)\big] \leq \frac{1}{\binom{n}{m}} \sum_{j \in \mathcal{J}} \mathbb{E}\big[\mathbb{W}(P_{W|S}, P_{W|S_{j^c}})\big],$$

where the expectation with respect to $P_J$ has been written explicitly. Then, this result follows by noting that, for all $i \in [n]$ and all $j \subseteq [n]$ such that $i \in j$

$$\mathbb{E}\big[\mathbb{W}(P_{W|Z_i}, P_W)\big] \leq \mathbb{E}\big[\mathbb{W}(P_{W|S}, P_{W|S_{j^c}})\big], \tag{9}$$

which is a stronger statement than the original. This is stronger since one can, without loss of generality, consider the samples $Z_i$ ordered so that the sequence $\{\mathbb{E}[\mathbb{W}(P_{W|Z_i}, P_W)]\}_{i \in [n]}$ is decreasing. Then, $\mathbb{E}[\mathbb{W}(P_{W|Z_1}, P_W)]$ is smaller than $\mathbb{E}[\mathbb{W}(P_{W|S}, P_{W|S_{j^c}})]$ for the $\binom{n-1}{m-1}$ sets $j$ in which sample 1 appears, $\mathbb{E}[\mathbb{W}(P_{W|Z_2}, P_W)]$ is smaller than $\mathbb{E}[\mathbb{W}(P_{W|S}, P_{W|S_{j^c}})]$ for the sets $j$ in which sample 2 appears and sample 1 does not, and so on.

More precisely, it can be shown that, for all $i \in [n]$ and all $j \subseteq [n]$ such that $i \in j$,

$$\mathbb{E}\left[ \sup_{f \in 1\text{-Lip}(\rho)} \Big\{ \mathbb{E}[f(W) \mid Z_i] - \mathbb{E}[f(W)] \Big\} \right] \leq \mathbb{E}\left[ \sup_{f \in 1\text{-Lip}(\rho)} \Big\{ \mathbb{E}[f(W) \mid S] - \mathbb{E}[f(W) \mid S_{j^c}] \Big\} \right],$$

which is equivalent to (9) due to the KR duality.

After writing $\mathbb{E}\big[ \sup_{f \in 1\text{-Lip}(\rho)} \big\{ \mathbb{E}[f(W) \mid S] - \mathbb{E}[f(W) \mid S_{j^c}] \big\} \big]$ in integral form, the result is shown as follows:

$$\int_{\mathcal{Z}^n} \sup_{f \in 1\text{-Lip}(\rho)} \left\{ \int_{\mathcal{W}} f(w) dP_{W|s}(w) - \int_{\mathcal{W}} f(w) dP_{W|s_{j^c}}(w) \right\} dP_S(s)$$

$$\overset{(a)}{\geq} \int_{\mathcal{Z}} \sup_{f \in 1\text{-Lip}(\rho)} \left\{ \int_{\mathcal{Z}^{n-1}} \left( \int_{\mathcal{W}} f(w) dP_{W|s}(w) - \int_{\mathcal{W}} f(w) dP_{W|s_{j^c}}(w) \right) P_Z^{\otimes n-1}(s^{-i}) \right\} dP_Z(z_i)$$

$$\overset{(b)}{=} \int_{\mathcal{Z}} \sup_{f \in 1\text{-Lip}(\rho)} \left\{ \int_{\mathcal{W}} f(w) dP_{W|z_i}(w) - \int_{\mathcal{W}} f(w) dP_W(w) \right\} dP_Z(z_i),$$

where $(a)$ stems from the fact that $\sup_g \mathbb{E}[g(X)] \leq \mathbb{E}[\sup_g g(X)]$, and $(b)$ follows from Fubini–Tonelli's theorem and the fact that $P_{W|Z_i} = \mathbb{E}[P_{W|S} \mid Z_i]$ and $P_W = \mathbb{E}[P_{W|S_{j^c}} \mid Z_i]$, since $i \in j$ and therefore $i \notin j^c$.

Finally, noting that $(b)$ is the integral form of $\mathbb{E}\big[ \sup_{f \in 1\text{-Lip}(\rho)} \big\{ \mathbb{E}[f(W) \mid Z_i] - \mathbb{E}[f(W)] \big\} \big]$ concludes the proof. $\square$

**Proposition 4.** *Consider the standard setting. Consider also a uniformly random subset of indices $J \subseteq [n]$ of size $m$. Then,*

$$\mathbb{E}\big[\mathbb{W}(P_{W|S}, P_{W|S_{J^c}})\big] \leq 2\mathbb{E}\big[\mathbb{W}(P_{W|S}, P_W)\big].$$

*Proof.* An application of the triangle inequality on Wasserstein distances [18, Chapter 6] states that, for all $j \subseteq [n]$ such that $|j| = m$ and all $s \in \mathcal{Z}^n$

$$\mathbb{W}(P_{W|s}, P_{W|s_{j^c}}) \leq \mathbb{W}(P_{W|s}, P_W) + \mathbb{W}(P_{W|s_{j^c}}, P_W). \tag{10}$$

Then, the inequality $\mathbb{E}[\mathbb{W}(P_{W|S_{j^c}}, P_W)] \leq \mathbb{E}[\mathbb{W}(P_{W|S}, P_W)]$ holds by the same arguments of Proposition 2. That is, writing the Wasserstein distance in its KR dual form and noting that the integral of a supremum is greater than the supremum of the integral and that $P_{W|S_{j^c}} = \mathbb{E}[P_{W|S} \mid S_{j^c}]$ for all $j \subseteq [n]$. Hence, taking expectations on both sides of (10) results in

$$\mathbb{E}\big[\mathbb{W}(P_{W|S}, P_{W|S_{J^c}})\big] \leq 2\mathbb{E}\big[\mathbb{W}(P_{W|S}, P_W)\big],$$

which completes the proof. $\square$

### D.1.2 Randomized-subsample setting

**Proposition 5.** *Consider the randomized-subsample setting. Then,*

$$\frac{1}{n}\sum_{i=1}^{n}\mathbb{E}\big[\mathbb{W}(P_{W|\tilde{S}_i,U_i}, P_{W|\tilde{S}_i})\big] \leq \mathbb{E}\big[\mathbb{W}(P_{W|\tilde{S},U}, P_{W|\tilde{S}})\big],$$

*where $\tilde{S}_i$ is defined in Theorem 3.*

*Proof.* The proposition follows noting that, for all $i \in [n]$

$$\mathbb{E}[\mathbb{W}(P_{W|\tilde{S}_i,U_i}, P_{W|\tilde{S}})] \leq \mathbb{E}[\mathbb{W}(P_{W|\tilde{S},U}, P_{W|\tilde{S}})], \tag{11}$$

which is a stronger statement than the original. Then, Equation (11) follows by the same arguments as Propositions 2 and 4. That is, writing the Wasserstein distance in its KR dual form and noting that the integral of a supremum is greater than the supremum of the integral and that $P_{W|\tilde{S}_i,U_i} = \mathbb{E}[P_{W|\tilde{S},U} \mid \tilde{S}_i, U_i]$ and $P_{W|\tilde{S}_i} = \mathbb{E}[P_{W|\tilde{S}} \mid \tilde{S}_i, U_i]$. $\square$

**Proposition 6.** *Consider the randomized-subsample setting. Consider also a uniformly random subset of indices $J \subseteq [n]$ of size $m$. Then,*

$$\frac{1}{n}\sum_{i=1}^{n}\mathbb{E}\big[\mathbb{W}(P_{W|\tilde{S}_i,U_i}, P_{W|\tilde{S}_i})\big] \leq \mathbb{E}\big[\mathbb{W}(P_{W|\tilde{S},U}, P_{W|\tilde{S},U_{J^c}})\big],$$

*Proof.* Note that the statement of the proposition may be written as

$$\frac{1}{n}\sum_{i=1}^{n}\mathbb{E}\big[\mathbb{W}(P_{W|\tilde{S}_i,U_i}, P_{W|\tilde{S}_i})\big] \leq \frac{1}{\binom{n}{m}}\sum_{j\in\mathcal{J}}\mathbb{E}\big[\mathbb{W}(P_{W|\tilde{S},U}, P_{W|\tilde{S},U_{j^c}})\big],$$

where the expectation with respect to $P_J$ has been written explicitly. Then, this result follows by noting that, for all $i \in [n]$ and all $j \subseteq [n]$ such that $i \in j$

$$\mathbb{E}\big[\mathbb{W}(P_{W|\tilde{S}_i,U_i}, P_{W|\tilde{S}_i})\big] \leq \mathbb{E}\big[\mathbb{W}(P_{W|\tilde{S},U}, P_{W|\tilde{S},U_{j^c}})\big], \tag{12}$$

which is a stronger statement than the original. Similarly to Proposition 4, this is stronger since one can, without loss of generality, consider the tuple of pairs of samples and their deciding index $(\tilde{S}_i, U_i) = \big((\tilde{Z}_i, \tilde{Z}_{i+n}), U_i\big)$ ordered so that the sequence $\{\mathbb{E}[\mathbb{W}(P_{W|\tilde{S}_i,U_i}, P_{W|\tilde{S}_i})]\}_{i\in[n]}$ is decreasing. Then, $\mathbb{E}[\mathbb{W}(P_{W|\tilde{S}_1,U_1}, P_{W|\tilde{S}_1})]$ is smaller than $\mathbb{E}[\mathbb{W}(P_{W|\tilde{S},U}, P_{W|\tilde{S},U_{j^c}})]$ for the $\binom{n-1}{m-1}$ sets $j$ in which the tuple 1 appears, $\mathbb{E}[\mathbb{W}(P_{W|\tilde{S}_2,U_2}, P_{W|\tilde{S}_2})]$ is smaller than $\mathbb{E}[\mathbb{W}(P_{W|\tilde{S},U}, P_{W|\tilde{S},U_{j^c}})]$ for the sets $j$ in which tuple 2 appears and tuple 1 does not, and so on.

Then, Equation (12) follows by the same arguments than Propositions 2 and 4. That is, writing the Wasserstein distance in its KR dual form and noting that the integral of a supremum is greater than the supremum of the integral and that $P_{W|\tilde{S}_i,U_i} = \mathbb{E}[P_{W|\tilde{S},U} \mid \tilde{S}_i, U_i]$ and $P_{W|\tilde{S}_i} = \mathbb{E}[P_{W|\tilde{S},U_{j^c}} \mid \tilde{S}_i, U_i]$, since $i \in j$ and therefore $i \notin j^c$. $\square$

**Proposition 7.** *Consider the randomized-subsample setting. Consider also a uniformly random subset of indices $J \subseteq [n]$ of size $m$. Then,*

$$\mathbb{E}\big[\mathbb{W}(P_{W|\tilde{S},U}, P_{W|\tilde{S},U_{J^c}})\big] \leq 2\mathbb{E}\big[\mathbb{W}(P_{W|\tilde{S},U}, P_{W|\tilde{S}})\big].$$

*Proof.* An application of the triangle inequality on Wasserstein distances [18, Chapter 6] states that, for all $j \subseteq [n]$ such that $|j| = m$, all $\tilde{s} \in \mathcal{Z}^{2n}$, and all $u \in [0,1]^n$

$$\mathbb{W}(P_{W|\tilde{s},u}, P_{W|\tilde{s},u_{j^c}}) \leq \mathbb{W}(P_{W|\tilde{s},u}, P_{W|\tilde{s}}) + \mathbb{W}(P_{W|\tilde{s},u_{j^c}}, P_{W|\tilde{s}}). \tag{13}$$

Then, the inequality $\mathbb{E}[\mathbb{W}(P_{W|\tilde{s},U_{j^c}}, P_{W|\tilde{s}})] \leq \mathbb{E}[\mathbb{W}(P_{W|\tilde{S},U}, P_{W|\tilde{s}})]$ holds by the same arguments of Proposition 2. That is, writing the Wasserstein distance in its KR dual form and noting that the integral of a supremum is greater than the supremum of the integral and that $P_{W|\tilde{s},U_{j^c}} = \mathbb{E}[P_{W|\tilde{S},U} \mid \tilde{S}, U_{j^c}]$ for all $j \subseteq [n]$. Hence, taking expectations on both sides of (13) results in

$$\mathbb{E}\big[\mathbb{W}(P_{W|\tilde{S},U}, P_{W|\tilde{S},U_{J^c}})\big] \leq 2\mathbb{E}\big[\mathbb{W}(P_{W|\tilde{S},U}, P_{W|\tilde{S}})\big],$$

which completes the proof. $\square$

### D.1.3 Comparison between the settings

**Proposition 8.** *Consider the standard and the randomized-subsample settings. Consider also a uniformly random subset of indices $J \subseteq [n]$ of size $m$. Then,*

$$\mathbb{E}[\mathbb{W}(P_{W|\tilde{S},U}, P_{W|\tilde{s}})] \leq 2\mathbb{E}[\mathbb{W}(P_{W|S}, P_W)],$$
$$\mathbb{E}[\mathbb{W}(P_{W|\tilde{S}_i,U_i}, P_{W|\tilde{S}_i})] \leq 2\mathbb{E}[\mathbb{W}(P_{W|Z_i}, P_W)], \text{ and}$$
$$\mathbb{E}[\mathbb{W}(P_{W|\tilde{S},U}, P_{W|\tilde{S},U_{J^c}})] \leq 2\mathbb{E}[\mathbb{W}(P_{W|S}, P_{W|S_{J^c}})].$$

*Proof.* The proofs of the three statements are analogous. Therefore only the proof of the first statement is explicitly written.

An application of the triangle inequality on Wasserstein distances [18, Chapter 6] states that, for $\tilde{s} \in \hat{\mathcal{Z}}^{2n}$ and all $u \in [0,1]^n$ such that $s$ is obtained through $\tilde{s}$ and $u$ as explained in the introduction,

$$\mathbb{W}(P_{W|\tilde{s},u}, P_{W|\tilde{s}}) \leq \mathbb{W}(P_{W|\tilde{s},u}, P_W) + \mathbb{W}(P_{W|\tilde{s}}, P_W). \tag{14}$$

Then, the inequality $\mathbb{E}[\mathbb{W}(P_{W|\tilde{S}}, P_W)] \leq \mathbb{E}[\mathbb{W}(P_{W|\tilde{S},U}, P_W)]$ holds by the same arguments of Proposition 2. That is, writing the Wasserstein distance in its KR dual form and noting that the integral of a supremum is greater than the supremum of the integral and that $P_{W|\tilde{S}} = \mathbb{E}[P_{W|\tilde{S},U} \mid \tilde{S}]$. Hence, taking expectations on both sides of (14) and noting that $P_{\tilde{S},U} = P_S$ almost surely results in

$$\mathbb{E}[\mathbb{W}(P_{W|\tilde{S},U}, P_{W|\tilde{s}})] \leq 2\mathbb{E}[\mathbb{W}(P_{W|S}, P_W)],$$

which completes the proof. $\square$

### D.2 Comparison of the mutual information based bounds

### D.2.1 Standard setting

In the standard setting, after a further application of Jensen's inequality, the bounds derived from Corollary 1 or [6, Proposition 1] are the tightest, followed by [4, Theorem 1] and the bounds derived from Corollary 2 when $|J| = 1$ or [7, Theorem 2.4] if the arbitrary random variable $R$ is not considered. This follows since by [6, Proposition 2] and [38, Lemma 3.7] or [9, Lemma 2]

$$\sum_{i=1}^{n} I(W; Z_i) \leq I(W; S) \leq \sum_{i=1}^{n} I(W; Z_i | S^{-i}),$$

where $S^{-i} = S \setminus Z_i$. More generally, with trivial modifications to the proof from [9, Lemma 2], it can be shown that

$$\sum_{i=1}^{n} I(W; Z_i) \leq I(W; S) \leq \mathbb{E}[I(W; S_J | S_{J^c})],$$

noting that, for any $j \subseteq [n]$ of size $m$, if the elements of $s_j$ are ordered as $Z_{k_1}, \ldots, Z_{k_m}$,

$$I(W; S) = \sum_{i=1}^{n} I(W; Z_i | S^{i-1}) \leq \frac{1}{\binom{n}{m}} \sum_{\iota=1}^{m} I(W; Z_{k_\iota} | S_{j^c}, S_j^{k_\iota - 1}),$$

since every time that $i = k_\iota$ then $S^{i-1} \subseteq (S_{j^c} \cup S_j^{k_\iota - 1})$, where $S_j^{k_\iota - 1}$ are the first $\iota$-1 elements of $S_j$.

### D.2.2 Randomized-subsample setting

With a similar argument to the one for the standard setting, after a further application of Jensen's inequality, the bounds derived from [10, Theorem 3.4] are tighter than [1, Theorem 5.1], which in turn are tighter than the bounds derived from Corollary 4 when $|J| = 1$ or [10, Theorem 3.7] if the arbitrary random variable $R$ is not considered, since

$$\sum_{i=1}^{n} I(W; U_i | \tilde{S}) \leq I(W; U | \tilde{S}) \leq \mathbb{E}[I(W; U_J | \tilde{S}, U_{J^c})].$$

Furthermore, the bounds derived from Corollary 3 or [9, Proposition 3] are the tightest as dictated by [9, Lemma 3] or [12, Lemma 2]. This way, the relationship between the conditional mutual information terms is

$$\sum_{i=1}^{n} I(W; U_i | \tilde{Z}_i, \tilde{Z}_{i+n}) \leq \sum_{i=1}^{n} I(W; U_i | \tilde{S}) \leq I(W; U | \tilde{S}) \leq \mathbb{E}[I(W; U_J | \tilde{S}, U_{J^c})].$$

### D.2.3 Comparison between the settings

Similarly, one may note that $I(W; U | \tilde{S}) \leq I(W; S)$ by [10], $I(W; U_i | \tilde{Z}_i, \tilde{Z}_{i+n}) \leq I(W; Z_i)$, and $I(W; U_j | \tilde{S}) \leq I(W; U_j | \tilde{S}, U_{j^c}) \leq I(W; S_j | S_{j^c})$ for any subset of indices $j \subseteq [n]$. Nonetheless, the additional factor of two in the bounds of the randomized-subsample setting makes the comparison between the bounds of the different settings harder.

An attempt for this comparison is given in [8], where they note that, since $(U, \tilde{S}) \leftrightarrow S \leftrightarrow W$ form a Markov chain and $S$ is a deterministic function of $\tilde{S}$ and $U$, then $I(W; S) = I(W; \tilde{S}) + I(W; U | \tilde{S})$, and hence the bound from the randomized-subsample setting [1, Theorem 5.1] is tighter than the one from the standard setting [4, Theorem 1] if $3I(W; U | \tilde{S}) \leq I(W; \tilde{S})$. There are similar requirements for the single-letter and random-subset bounds, namely:

- The bound derived from Corollary 3 or [9, Proposition 3] is tighter than [6, Proposition 1] if $3I(W; U_i | \tilde{Z}_i, \tilde{Z}_{i+n}) \leq I(W; \tilde{Z}_i, \tilde{Z}_{i+n})$.

- The bound derived from Corollary 4 or [10, Theorem 3.7] is tighter than [7, Theorem 2.4] if $3I(W; U_j | \tilde{S}, U_{j^c}) \leq I(W; \tilde{S}_j | S_{j^c})$.

**Remark 4.** *Note that, sometimes, seemingly looser bounds can lead to tighter or more tractable bounds for specific algorithms. For instance, the random-subset bounds from [7, 9, 10] lead to tighter Langevin dynamics and stochastic gradient Langevin dynamics than the single-letter bounds from [6].*

## E  Derivations for the Gaussian location model example

The problem considered in the example is the estimation of the mean $\mu$ of a $d$-dimensional Gaussian distribution with known covariance matrix $\sigma^2 I_d$. Furthermore, there are $n$ samples $S = (Z_1, \ldots, Z_n)$ available, the loss is measured with the Euclidean distance $\ell(w, z) = \|w - z\|_2$, and the estimation is their empirical mean $W = \frac{1}{n} \sum_{i=1}^{n} Z_i$.

To calculate the expected generalization error and derive different bounds, it is convenient to know how the random variables are distributed. For example, in this setting $P_Z = \mathcal{N}(\mu, \sigma^2 I_d)$, $P_W = \mathcal{N}\left(\mu, \frac{\sigma^2}{n} I_d\right)$, $P_{W|Z_i} = \mathcal{N}\left(\frac{(n-1)\mu + Z_i}{n}, \frac{\sigma^2(n+1)}{n^2} I_d\right)$, $P_{W|S^{-j}} = \mathcal{N}\left(\frac{\mu}{n} + \frac{1}{n} \sum_{i \neq j} Z_i, \sigma^2 I_d\right)$, and $P_{W|S} = \delta\left(\frac{1}{n} \sum_{i=1}^{n} Z_i\right)$. Another important feature of this problem is that the loss function is 1-Lipschitz under $\rho(w, w') = \|w - w'\|_2$.

### E.1  Expected generalization error

In order to derive an exact expression of the generalization error, it is suitable to write it in the following explicit form:

$$\overline{\text{gen}}(W, S) = \mathbb{E}[\ell(W, Z)] - \frac{1}{n} \sum_{i=1}^{n} \mathbb{E}[\ell(W, Z_i)],$$

where $Z \sim P_Z$ is independent of $W$. Then, the two terms can be evaluated independently.

The first term is equivalent to

$$\mathbb{E}[\ell(W, Z)] = \mathbb{E}[\|W - Z\|_2] = \sqrt{2\sigma^2 \left(1 + \frac{1}{n}\right)} \frac{\Gamma\left(\frac{d+1}{2}\right)}{\Gamma\left(\frac{d}{2}\right)},$$

where the first equality follows from the definition of the loss function. The second equality follows from noting that $(W - Z) \sim \mathcal{N}\big(0, \sigma^2\big(1 + \frac{1}{n}\big)I_d\big)$ and therefore $\|W - Z\|_2 = \sqrt{\sigma^2\big(1 + \frac{1}{n}\big)}X$, where $X$ is distributed according to the chi distribution with $d$ degrees of freedom.

Similarly, the summands of the second term are equivalent to

$$\mathbb{E}[\ell(W, Z_i)] = \mathbb{E}[\|W - Z_i\|_2] = \sqrt{2\sigma^2\Big(1 - \frac{1}{n}\Big)}\frac{\Gamma\big(\frac{d+1}{2}\big)}{\Gamma\big(\frac{d}{2}\big)},$$

where as before the first equality follows from the definition of the loss function. The second equality follows from noting that $(W - Z_i) \sim \mathcal{N}\big(0, \sigma^2\big(1 - \frac{1}{n}\big)I_d\big)$ and therefore $\|W - Z_i\|_2 = \sqrt{\sigma^2\big(1 - \frac{1}{n}\big)}X$, where $X$ is distributed according to the chi distribution with $d$ degrees of freedom. In this case, $W$ and $Z_i$ are not independent random variables. In fact, $(W, Z_i)$ is normally distributed with covariance matrix

$$\begin{pmatrix} \frac{\sigma^2}{n}I_d & \frac{\sigma^2}{n}I_d \\ \frac{\sigma^2}{n}I_d & \sigma^2 I_d \end{pmatrix},$$

from which the distribution of $W - Z_i$ is deduced.

Finally, subtracting both terms results in

$$\overline{\mathrm{gen}}(W, S) = \sqrt{\frac{2\sigma^2}{n}}\big(\sqrt{n+1} - \sqrt{n-1}\big)\frac{\Gamma\big(\frac{d+1}{2}\big)}{\Gamma\big(\frac{d}{2}\big)} \leq \frac{\sqrt{2\sigma^2 d}}{n}.$$

where the inequality follows from the following two bounds: (i) $\sqrt{n+1} - \sqrt{n-1} \leq \sqrt{\frac{2}{n}}$, which is obtained by multiplying and dividing by $\sqrt{n+1}+\sqrt{n-1}$ and noting that $\sqrt{n+1}+\sqrt{n-1} \geq \sqrt{2n}$, and (ii) the upper bound on the ratio of gamma distributions by $\sqrt{\frac{d}{2}}$ using the series expansion at $d \to \infty$.

## E.2   Wasserstein distance bound

The bound from [15] can be calculated exactly since $P_{W|S}$ is a delta distribution, that is

$$\mathbb{E}\big[\mathbb{W}(P_{W|S}, P_W)\big] = \mathbb{E}\bigg[\bigg\|\frac{1}{n}\sum_{i=1}^{n} Z_i - \frac{1}{n}\sum_{i=1}^{n} Z_i'\bigg\|_2\bigg] = \sqrt{\frac{4\sigma^2}{n}}\frac{\Gamma\big(\frac{d+1}{2}\big)}{\Gamma\big(\frac{d}{2}\big)} \leq \sqrt{\frac{2\sigma^2 d}{n}}$$

where $Z_i' \sim P_Z$ are independent copies of $Z_i$. Hence, the difference is distributed as a normal distribution with mean 0 and covariance $\frac{2\sigma^2}{n}I_d$, which means that the norm is $\sqrt{\frac{2\sigma^2}{n}}X$, where $X$ is a chi random variable with $d$ degrees of freedom.

## E.3   Individual sample Wasserstein distance bound

An exact calculation of the bound from Theorem 1 is cumbersome. However, the Wasserstein distance of order one can be bounded from above by the Wasserstein distance of order two (Remark 1), which has a closed form expression for Gaussian distributions. More specifically,

$$\mathbb{E}\big[\mathbb{W}(P_{W|Z_i}, P_W)\big] \leq \mathbb{E}\big[\mathbb{W}_2(P_{W|Z_i}, P_W)\big] \leq \frac{\sqrt{2\sigma^2}}{n}\frac{\Gamma\big(\frac{d+1}{2}\big)}{\Gamma\big(\frac{d}{2}\big)} + \sqrt{\frac{\sigma^2 d}{n^3}} \leq \frac{\sqrt{\sigma^2 d}}{n} + \sqrt{\frac{\sigma^2 d}{n^3}}.$$

The second inequality follows from the closed-form expression for the squared Wasserstein distance of order 2, namely

$$\mathbb{W}(P_{W|Z_i}, P_W)^2 = \frac{1}{n^2}\|\mu - Z_i\|^2 + \frac{\sigma^2 d}{n}\Big(1 + \frac{n-1}{n} - 2\sqrt{\frac{n-1}{n}}\Big),$$

where the term $\big(1 + \frac{n-1}{n} - 2\sqrt{\frac{n-1}{n}}\big)$ is a perfect square that is bounded from above by $\frac{1}{n^2}$. Then the expression results from employing the inequality $\sqrt{x + y} \leq \sqrt{x} + \sqrt{y}$ and noting that $\|\mu - Z_i\|_2 = \sigma X$, where $X$ is a chi distributed random variable with $d$ degrees of freedom.

### E.4 Random subset Wasserstein distance bound

As in E.2, since $P_{W|S}$ is a delta distribution, the bound from Theorem 2 can be calculated exactly. In particular, the bound assuming that $|J| = 1$ is

$$
\mathbb{E}\big[\mathbb{W}(P_{W|S}, P_{W|S^{-J}})\big] = \mathbb{E}\Big[\Big\|\frac{1}{n}\sum_{i=1}^{n} Z_i - \Big(\frac{Z_J'}{n} + \frac{1}{n}\sum_{i\neq J} Z_i\Big)\Big\|_2\Big] = \frac{\sqrt{4\sigma^2}}{n}\frac{\Gamma\big(\frac{d+1}{2}\big)}{\Gamma\big(\frac{d}{2}\big)} \leq \frac{\sqrt{2\sigma^2 d}}{n},
$$

where $Z_J' \sim P_Z$ is an independent copy of $Z_J$. Hence the norm is $\frac{\sqrt{2\sigma^2}}{n}X$, where $X$ is a chi random variable with $d$ degrees of freedom.

### E.5 Individual sample mutual information bound

The individual sample mutual information is $I(W; Z_i) = \frac{d}{2}\log\big(\frac{n}{n-1}\big)$ for all $i \in [n]$ [6]. Nonetheless, in order to employ the bound from [6], the loss function $\ell(W, Z)$ needs to have a cumulant generating function $\Lambda(\lambda)$ bounded from above by a convex function $\psi(\lambda)$ such that $\psi(0) = \psi'(0) = 0$ for all $\lambda \in (-b, 0]$ for some $b \in \mathbb{R}_+$, where $Z \sim P_Z$ is independent of $W$.

The loss function $\ell(W, Z) = \|W - Z\|_2$ is $\sqrt{\sigma^2(1 + \frac{1}{n})}X$, where $X$ is a chi random variable with $d$ degrees of freedom. The moment generating function $M(\lambda)$ of such a random variable is

$$
M(\lambda) = \bar{M}\Big(\frac{d}{2}, \frac{1}{2}, \frac{\lambda^2}{2}\Big) + \frac{\lambda\sqrt{2}\Gamma\big(\frac{d+1}{2}\big)}{\Gamma\big(\frac{d}{2}\big)}\bar{M}\Big(\frac{k+1}{2}, \frac{3}{2}, \frac{\lambda^2}{2}\Big),
$$

where $\bar{M}$ is the Kummer's confluent hypergeometric function.

The expression of this moment generating function is too convoluted to study for $d > 1$. Nonetheless, for $d = 1$ it has a closed form expression, namely

$$
M(\lambda) = e^{\frac{\lambda^2}{2}}\Big(1 + \mathrm{erf}\Big(\frac{\lambda}{\sqrt{2}}\Big)\Big).
$$

Therefore, the cumulant generating function is $\Lambda(\lambda) = \frac{\lambda^2}{2} + \log(1 + \mathrm{erf}(\frac{\lambda}{\sqrt{2}}))$, which is bounded from above by the convex function $\psi(\lambda) = \frac{\lambda^2}{2}$ for all $\lambda \in (-\infty, 0]$. Hence, the bound from [6] can be applied yielding

$$
\overline{\mathrm{gen}}(W, S) \leq \frac{1}{n}\sum_{i=1}^{n}\sqrt{2\sigma^2\Big(1 + \frac{1}{n}\Big)I(W; Z_i)} \leq \sqrt{\sigma^2\Big(1 + \frac{1}{n}\Big)\log\Big(\frac{n}{n-1}\Big)} \leq \sqrt{\frac{2\sigma^2}{n-1}},
$$

where the last inequality stems from noting that $\frac{n}{n-1} = 1 + \frac{1}{n-1}$, the fact that $\log(1 + x) \leq x$, and bounding $(1 + \frac{1}{n})$ from above by 2.

## F  Randomized-subsample setting and the BH inequality

In Corollaries 3 and 4, the immediate bound that stems from the use of the BH inequality is not included. The reason for this is that the relative entropies $D_{\mathrm{KL}}(P_{W|\bar{Z}_i, \bar{Z}_{i+n}, U_i} \| P_{W|\bar{Z}_i, \bar{Z}_{i+n}})$ and $D_{\mathrm{KL}}(P_{W|\bar{S}, U, R} \| P_{W|\bar{S}, U_{J^c}, R})$, when $|J| = 1$, are never greater than $\log(2)$ as shown in Lemma 3 below. Hence, the range of these relative entropies is inside the range where Pinsker's inequality is tighter than the BH inequality.

**Lemma 3.** *Let $P_{X|A,B}$ be a conditional probability distribution on $\mathcal{X}$, where $B$ is a Bernoulli random variable with probability $1/2$ and $A \in \mathcal{A}$ is a random variable independent of $B$. Let also $P_{X|A} = \mathbb{E}[P_{X|A,B} \mid A]$. Then, $D_{\mathrm{KL}}(P_{X|A,B} \| P_{X|A}) \leq \log(2)$.*

*Proof.* In this situation $P_{X|A}$ dominates $P_{X|A,B}$ and $P_{X|A,(1-B)}$, that is $P_{X|A,B} \ll P_{X|A}$ and $P_{X|A,(1-B)} \ll P_{X|A}$, since $P_{X|A} = (P_{X|A,B} + P_{X|A,(1-B)})/2$. Therefore,

$$
\begin{aligned}
D_{\mathrm{KL}}(P_{X|A,B} \parallel P_{X|A}) &= \mathbb{E}\left[ \log \left( \frac{dP_{X|A,B}}{d\left(\frac{1}{2}P_{X|A,B} + \frac{1}{2}P_{X|A,(1-B)}\right)} \right) \middle| A, B \right] \\
&\overset{(a)}{=} -\mathbb{E}\left[ \log \left( \frac{d\left(\frac{1}{2}P_{X|A,B} + \frac{1}{2}P_{X|A,(1-B)}\right)}{dP_{X|A,B}} \right) \middle| A, B \right] \\
&\overset{(b)}{=} \log(2) - \mathbb{E}\left[ \log \left( 1 + \frac{dP_{X|A,(1-B)}}{dP_{X|A,B}} \right) \middle| A, B \right] \\
&\overset{(c)}{\leq} \log(2),
\end{aligned}
$$

where $(a)$ stems from [39, Exercise 9.27], $(b)$ follows from the linearity $P_{X|A,B}$-a.e. of the Radon–Nikodym derivative, and $(c)$ is due to the fact that $\log(1 + x) \geq 0$ for all $x \geq 0$ and the fact that $dP_{X|A,(1-B)}/dP_{X|A,B}$ is always positive. Also, steps $(a)$, $(b)$, and $(c)$ are possible since the expectation integrates over the support of $P_{X|A,B}$, avoiding the problems of absolute continuity in $(c)$ and absorbing the $P_{X|A,B}$-a.e. properties for $(a)$ and $(b)$. $\square$

**Remark 5.** *Lemma 3 can be easily extended to the case where $B$ is a sequence of $k$ Bernoulli random variables $B_i$, noting that $P_{X|A} = 2^{-k} \sum_{j=1}^{2^k} P_{X|A,\mathcal{B}_j}$, where $\mathcal{B}$ are all the $2^k$ random sequences $\mathcal{B}_j$ where the $i$-th element can be either $B_i$ or $(1 - B_i)$. In that case, we have that $D_{\mathrm{KL}}(P_{X|A,B} \parallel P_{X|A}) \leq k \log(2)$.*

Then, note that $D_{\mathrm{KL}}(P_{W|\tilde{Z}_i,\tilde{Z}_{i+n},U_i} \parallel P_{W|\tilde{Z}_i,\tilde{Z}_{i+n}}) \leq \log(2)$ if $A = (\tilde{Z}_i, \tilde{Z}_{i+n})$, $B = U_i$, and $X = W$. Similarly, for $|J| = 1$, note that $D_{\mathrm{KL}}(P_{W|\tilde{S},U,R} \parallel P_{W|\tilde{S},U_{J^c},R}) \leq \log(2)$ if $A = (\tilde{S}, U_{J^c}, R)$, $B = U_J$, and $X = W$.

**Remark 6.** *In Corollary 4, when $|J| > 2$, it is not guaranteed that the inequality obtained from Pinsker's inequality is tighter than the one obtained with the BH inequality. For instance, as per Remark 5, $D_{\mathrm{KL}}(P_{W|\tilde{S},U,R} \parallel P_{W|\tilde{S},U_{J^c},R})$ could be as large as $|J|\log(2)$, which is already larger than $1.6$ for $|J| = 3$. Hence, for $|J| > 2$, one should also consider that inequality if one desires the tightest bound.*

*However, the bound derived from the BH inequality was not included since this kind of bounds are usually employed for $|J| = 1$, e.g., [10, Theorem 4.2] and [9, Proposition 6]. Moreover, after applying Jensen's inequality, it is shown that the derived mutual-information bounds are the tightest when $|J| = 1$ [10, Corollary 3.3].*

## G Rate-distortion theory and generalization

### G.1 Rate–distortion theory

Rate–distortion theory [22, 19] deals with the problem of determining the minimum number of bits, determined by the *rate* $R$, that should be employed to characterize a signal $X$ by $Y$ so that this signal can later be recovered with an expected *distortion* lower than $\delta$. Formally, given a signal $X$ with distribution $P_X$ and a distortion measure $d$, the *rate–distortion* function $R(\delta)$ finds the optimal encoding distribution $P_{Y|X}^\star$, i.e., the channel $P_{Y|X}$ that generates a representation $Y$ with the minimum amount of bits $R(\delta)$ and an expected distortion lower than $\delta$. Namely,

$$
R(\delta) = \inf_{P_{Y|X} : \mathbb{E}[d(X,Y)] \leq \delta} I(X;Y).
$$

A celebrated result in rate–distortion theory is its duality. More precisely, instead of looking for the channel $P_{Y|X}$ that most compresses a signal $X$ with a limited distortion $\delta$, one can look for the channel $P_{Y|X}$ that less distorts the signal $X$ with a limited budget of bits $r$. That is, one can solve the *distortion–rate* function

$$
D(r) = \inf_{P_{Y|X} : I(X;Y) \leq r} \mathbb{E}[d(X,Y)].
$$

Then, the duality theorem states that $R(\delta) = D^{-1}(\delta)$ and $D(r) = R^{-1}(r)$ [40, Lemma 4.1.2], or, in words, that the inverse of the rate–distortion function is the distortion–rate function, and vice versa.

**Remark 7.** *Rate–distortion theory is a well-studied field and the rate–distortion and distortion–rate functions have many more interesting properties and analytical solutions and bounds for particular (and common) cases.*

**Single-letterization**  Sometimes, if $X$ is a sequence of signals $X = (X_1, \ldots, X_n)$, solving the rate–distortion function is challenging. Assume that the signals $X_i$ are independent and the distortion $d$ is separable, i.e., $\mathbb{E}[d(X, Y)] = \frac{1}{n} \sum_{i=1}^{n} \mathbb{E}[d(X_i, Y_i)]$. Then, a simpler task is to solve the single-letter version of the rate–distortion function. Namely, for any $i \in [n]$, to solve

$$R_i(\delta_i) = \inf_{P_{Y_i|X_i}:\mathbb{E}[d(X_i,Y_i)]\leq\delta} I(X_i; Y_i).$$

Then, for any $i$, the equality $nR_i(\delta) = R(\delta)$ [19, Theorem 26.1] holds. Hence, the channel $P_{Y|X} = P_{Y_i|X_i}^{\otimes n}$ can be used as a proxy for $P_{Y|X}^{\star}$.

**Backward-channel**  Sometimes, working with the backward channel $P_{X|Y}$ is convenient to derive an analytical solution to the rate–distortion function for a particular input signal distribution, e.g., Bernoulli or Gaussian [19, Chapter 27.1]; to derive an analytical bound for certain distribution families, e.g., bounded variance; or to derive an analytical bound for certain distortion families, e.g., difference, additive, or autoregressive distortions [40, Sections 4.3, 4.6, and 4.7]

### G.2  Connection to the generalization error

When designing a learning algorithm $P_{W|S}$, the aim is an algorithm that attains a low accuracy error (or risk) while generalizing well. This informal sentiment can be posed as constrained optimization as follows: to design an algorithm $P_{W|S}$ that has the minimum possible expected generalization error $G(\epsilon)$ while maintaining a population risk lower than $\epsilon$. Namely, one may consider the *generalization–risk* function to be

$$G(\epsilon) = \inf_{P_{W|S}:\mathbb{E}[L_S(W)]\leq\epsilon} \mathbb{E}[\mathrm{gen}(W, S)],$$

and select the algorithm $P_{W|S}^{\star}$ that solves it.

Furthermore, the expected generalization error increases with $I(W; S)$ as shown in [4, Theorem 1] or (2). Therefore, one may instead consider the *information-generalization–risk* function

$$G^{\blacktriangle}(\epsilon) = \inf_{P_{W|S}:\mathbb{E}_{P_{W,S}}[L_S(W)]\leq\epsilon} I(W; S),$$

and use it as a surrogate of the generalization–risk function to choose the algorithm $P_{W|S}^{\blacktriangle}$ that solves $G^{\blacktriangle}(\epsilon)$ as a proxy for $P_{W|S}^{\star}$. Therefore, the powerful rate–distortion theory may be employed to select a sensible learning algorithm or, at least, to better understand the trade-off between generalization and risk.

**Remark 8.** *There are several issues to be considered when applying rate–distortion theory to the information-generalization–risk function. For instance, usually a hypothesis is not separable, i.e., $W \neq (W_1, \ldots, W_n)$, and hence the single-letterization of the problem must be carried obtaining $P_{W|Z_i}$ instead of $P_{W_i|Z_i}$, which is inconvenient since then obtaining $P_{W|S}$ from $P_{W|Z_i}$ is cumbersome. Nonetheless, the aim of this section is just to give some intuition about the connection between rate–distortion and generalization, and a proper formal framework is beyond the objective of this paper. The reader is referred to [11] and [23] for other connections between these two concepts.*

## H  Additional remarks on the chi-squared based bounds

As mentioned in §3.4, the presented bounds in terms of the total variation result in bounds based on the $\chi^2$-divergence and other $f$-divergences employing the joint range strategy [19, Chapter 7]. In order to do so, the loss function $\ell$ is required to be $L$-Lipschitz for all $z \in \mathcal{Z}$ under the discrete metric, or, in other words, to be bounded in a range $[c, c + L]$ for some $c \in \mathbb{R}$.

**Claim 1.** *If a function $f$ is L-Lipschitz under the discrete metric, it is bounded in $[c, c + L]$ for some $c \in \mathbb{R}$.*

*Proof.* If $|f(x) - f(y)| \leq L\rho_H(x, y)$ for all $x, y \in \mathcal{X}$ then $|f(x) - f(y)| \leq L$. This holds if and only if $f : \mathcal{X} \to [c, c + L]$ for some $c \in \mathbb{R}$. $\qquad\square$

As an example, Equation (6) is obtained as a corollary of Theorem 1. However, note that Theorems 1, 2, 3, and 4, and Equations (1) and (3) can be replicated using the variational representation of the $\chi^2$-divergence [41, Example 6.4], which states that for all distributions $P, Q$ over $\mathcal{X}$

$$\chi^2(P, Q) = \sup_{f:\mathcal{X}\to\mathbb{R}} \left\{ \frac{\left(\mathbb{E}[f(X)] - \mathbb{E}[f(X)]\right)^2}{\mathrm{Var}[Y]} \right\},$$

where $X \sim P, Y \sim Q$, and the supremum is taken over all functions $f$ with finite expectation with respect to $P$ and $Q$ and finite variance with respect to $Q$. Using this tool instead of the KR duality, Theorem 1 would result in

$$\left|\overline{\mathrm{gen}}(W, S)\right| \leq \frac{1}{n} \sum_{i=1}^{n} \mathbb{E}\left[ \sqrt{\mathrm{Var}\left[\ell(W', Z_i)\right] \chi^2(P_{W|Z_i}, P_W)} \right], \tag{15}$$

where $W'$ is an independent copy of $W$ such that $P_{W,Z_i} = P_W \otimes P_{Z_i}$ for all $i \in [n]$. Equation (15), instead of requiring that the loss $\ell$ is L-Lipschitz under the discrete metric for all $z \in \mathcal{Z}$, it requires that the function $\ell(w, z)$ has finite expectation with respect to $P_{W|Z_i=z}$ and $P_W$ and finite variance with respect to $P_W$ for all $z \in \mathcal{Z}$. Note that this is a weaker requirement since Lipschitzness under the discrete metric implies boundedness (Claim 1) and Popoviciu's inequality states that a bounded random variable has finite variance.

In particular, under the assumption that $\ell$ is bounded with a range $[c, L + c]$, one may use Popoviciu's inequality [42], i.e., $\mathrm{Var}[\ell(W', Z_i)] \leq L^2/4$, to recover the corollaries of Theorems 1, 2, 3, and 4 using the discrete metric and the joint range. As an example, using Popoviciu's inequality in combination with (15) recovers (6). Hence, the bounds obtained through the variational representation of the $\chi^2$-divergence are both tighter and more general than those obtained after applying the joint range strategy as in [43, §IV-B] to the total variation bounds obtained through the KR duality. Compared to the bound from (5), obtained after applying first Pinsker's inequality and then the joint range strategy, Equation (15) becomes looser as soon as

$$\chi^2(P_{W|Z_i}, P_W) \geq \frac{-1}{\alpha} \left( \mathtt{W}\left( - \alpha e^{-\alpha} \right) + \alpha \right),$$

where $\mathtt{W}$ denotes the Lambert or product-log function and $\alpha = \mathrm{Var}[\ell(W', Z_i)]/2$. In the extreme case where $\mathrm{Var}[\ell(W', Z_i)] = L^2/4$, Equation (5) is tighter than (15) as soon as $\chi^2(P_{W|Z_i}, P_W) \gtrsim 2.51$, a restriction that becomes more favorable to (15) as the variance decreases. More precisely, the bound obtained from the variational representation of the $\chi^2$-divergence is tighter than (5) in the range where both bounds are non-vacuous, i.e., when $\mathrm{Var}[\ell(W', Z_i)] \leq (e^2 - 1)^{-1}L^2$.