# OpenReview forum: "Tighter Expected Generalization Error Bounds via Wasserstein Distance"
_NeurIPS.cc/2021/Conference — NeurIPS 2021 Poster_

### Official Review · Reviewer_SCNA · 2021-06-27

**Rating:** 7
**Confidence:** 5

**Summary:**

In this paper new generalization error bounds based on Wasserstein distance are derived. As shown in the paper, these bounds are tighter than existing bounds based on mutual information. Also, the authors briefly discuss how their bounds could be extended to deal with backward channels (instead of forward channels), and how they could lead to bounds based on general f-divergences.

**Limitations And Societal Impact:**

Yes, the authors adequately addressed the limitations of their work.

**Main Review:**

The literature review is very well done, which helps to put the paper contribution in perspective. While no new techniques are introduced, the authors made a good job in refining and combining different existing techniques. As far as the reviewer is concerned, it seems like the bounds presented in this paper are close to the best that can be done with existing techniques at the proposed level of generality. Applying these bounds beyond toy examples still requires some effort (probably a big one), but the paper reveals a clear path to do so. Overall, this paper represents a solid step towards practical generalization bounds.

Minor Suggestions
- It would be helpful for the reader if the authors emphasize, perhaps in a graphical manner, the taxonomy of current generalization bounds in terms of performance.
- Bobkov-Götze's theorem has a similar role to the HB inequality, although the former is not presented in the preliminaries. Perhaps it is useful to add the BG theorem to the preliminaries.

**Time Spent Reviewing:**

10

---

> ### Author Response · Authors · 2021-08-09
> **Answer to the comments**
>
> Dear Reviewer SCNA,
>
> Thank you for taking the time to review and assess our paper. Please, find below our answer to your suggestions. We will first re-write your suggestions and will follow them with our answers.
>
> * **Reviewer**: It would be helpful for the reader if the authors emphasize, perhaps in a graphical manner, the taxonomy of current generalization bounds in terms of performance.
>
>     **Answer:** This is a fantastic idea, thanks. Unfortunately, we do not have space in the main text due to the page restrictions. Nonetheless, we will create a tikz figure (or figures) relating all these different bounds and include it in Appendix C.
>
> * **Reviewer:** The comparison of the Wasserstein distance-based bounds presented in Appendix C should be included in the main body of the paper to demonstrate the tightness of the proposed bound. It would also be helpful to construct a table which summarizes the comparisons between all these bounds.
>
>     **Answer:** As we mentioned to Reviewer Bm5Z, who also suggested including this comparison in the main text, this is also a great idea that will improve the reader experience, thank you. We now included an informal proposition right before the example summarizing the relationships between the different Wasserstein distance bounds.
>
>     As for the table, we believe that with the graphical figure you suggested the relationships between the different bounds will be clear and well organized.
>
> * **Reviewer**: Bobkov-Götze's theorem has a similar role to the HB inequality, although the former is not presented in the preliminaries. Perhaps it is useful to add the BG theorem to the preliminaries.
>
>     **Answer:** We agree that it would be good to include that in the main body. Nonetheless, due to page constraints (Appendix A.1 takes $\approx 3/4$ of a page) we decided to prioritize the exposition of the main ideas in the 9-pages main body and defer the generality provided by the Bobkov-Götze to the Appendix.

---

> ### Comment · Reviewer_SCNA · 2021-08-17
> **Post Rebbutal**
>
> Dear all,
>
> Since my score is the only one in disagreement, I feel obligated to further explain it.
>
> I agree that the authors do not address appropriately the issues on how to find a useful metric or how to design a tractable coupling (and the argument that they can recover other bounds is not satisfactory). However, it is not uncommon in the Wasserstein distance literature to solve that kind of problems on a case by case basis, as pointed out by the authors in their response to Reviewer 61bD. While these issues make the application of the proposed bounds not straightforward, I do believe that they are significant, as mentioned by Reviewer Bm5Z. In particular, the proposed bounds seem easier to implement than previous ones (e.g., those by Wang et al.) and, as evidenced by their toy example, they seem to remain tight despite their "simplicity".
>
> While there are non-trivial problems yet to solve, this paper seems to lay down a very promising framework to derive practical generalization bounds based on Wasserstein distance. I leaved my score unchanged.

---

### Official Review · Reviewer_Bm5Z · 2021-07-15

**Rating:** 6
**Confidence:** 4

**Summary:**

This work presents several expected generalization error bounds in terms of the Wasserstein distance. Several settings (full-dataset, single-letter, and random-subset) are considered in the paper. Under the assumption that the loss function is bounded and using Hamming distance in the Wasserstein distance, the proposed bounds are shown to be tighter than the existing mutual information (or KL divergence) based generalization error bound. The author also claims that serval new bounds based on a variety of information measures can be obtained using these Wasserstein distance-based bounds.



**Limitations And Societal Impact:**

Most limitations are well addressed in Section 3.5 and 4.1. Another limitation is that the Wasserstein distance is hard to evaluate and estimate in practice. I would like the authors to add more discussions to address this issue.

**Main Review:**

The paper is technically sound and well-written.

My major concern about this work is the novelty. The idea of bounding generalization error using Wasserstein distance is presented in [14] and [16] in the reference of the paper. And the techniques used in the paper to tighten the Wasserstein distance-based bounds are originally designed for tightening mutual information-based bounds, including single-letter characterization from [6], random-subset [9] and randomized subsample setting from [1].

However, the result obtained by combining those techniques with Wasserstein distance-based bounds is significant. Example 1 shows that the proposed method could provide an order optimal bound for the generalization error of Gaussian mean estimation problem, which seems to be the first order optimal information-theoretic bound based on my knowledge. It was shown in [6] that simply using individual mutual information I(W;Z_i) will result in a sub-optimal bound, so I would like the author to provide more explanation and discussion about this improvements. It seems that the improvement is related to the metric used in the Wasserstein distance. Also, it is the best to include more examples to demonstrate the powerfulness of the proposed bound.

Minor comments:
1.	The results presented in section 3.3 and 3.4 are quite straight forward, and it is hard for me to see the values of those results based on the discussion in the current paper. More discussion and some potential applications of these results would be helpful. Otherwise, there is no reason to include some looser bounds in the main body of the paper.
2.	The comparison of the Wasserstein distance-based bounds presented in Appendix C should be included in the main body of the paper to demonstrate the tightness of the proposed bound. It would also be helpful to construct a table which summarizes the comparisons between all these bounds.
3.	The ICIMI bound from [12] could be added in Figure 1 as another baseline.


**Time Spent Reviewing:**

3

---

> ### Author Response · Authors · 2021-08-09
> **Answer to the comments**
>
> Dear Reviewer Bm5Z,
>
> Thank you for reviewing and assessing our paper. We will respond to your remarks (in order) below. We will first re-write your comments/questions and will follow them with our answers.
>
> * **Reviewer**:  Example 1 shows that the proposed method could provide an order optimal bound for the generalization error of Gaussian mean estimation problem, which seems to be the first order optimal information-theoretic bound based on my knowledge. It was shown in [6] that simply using individual mutual information $I(W;Z_i)$ will result in a sub-optimal bound, so I would like the author to provide more explanation and discussion about this improvements. It seems that the improvement is related to the metric used in the Wasserstein distance.
>
>     **Answer**: The particular example of the Gaussian location model illustrates two benefits from bounds based on the Wasserstein distance compared to those based on the relative entropy and the mutual information.
>
>     * The pitfall of potentially infinite bounds is erased. The bound based on the mutual information $I(W;S)$ is infinite since $W$ is a deterministic function of $S$ and $S$ is a continuous random variable. Conversely, the Wasserstein distance $\mathbb{W}(P_{W|S},P_W) \in \mathcal{O}(\sqrt{\sigma^2 d / n})$.
>
>     * The bound is linear with the metric. In this particular example of comparing Gaussians, the mutual information of $W$ and $Z_i$ is quite good in capturing how far apart are the joint and product distributions of the random variables, it is actually rate-optimal: $I(W;Z_i) = \log \big(\frac{n}{n-1} \big) \in \mathcal{O}(1/n)$.
>         However, it has the pitfall that the bound is *not* linear on $I(W;Z_i)$, but on $\sqrt{I(W;Z_i)}$.
>         Nonetheless, the Wasserstein distance is also good at capturing the distance between the posterior and prior distributions of the hypothesis and it is also rate optimal: $\mathbb{W}(P_{W|{Z_i}},P_W) \in \mathcal{O}(1/n)$.
>         The advantage here is that the bound *is* linear with this metric.
>         Therefore, relative-ntropy--based bounds always have a square root disadvantage with respect to Wasserstein-distance--based bounds.
>
>     There are other situations where the usage of the metric could be more beneficial when compared to the relative entropy. For instance, for discrete distributions. Here, if the forward channel does not place some probability to all of the possible hypotheses, then the relative entropy explodes. Moreover, if there is no geometry to exploit such as in a categorical distribution and the loss is bounded, e.g. 0-1 loss, then employing the Wasserstein distance with the discrete metric (i.e. the total variation) will always give a tighter bound due to Pinsker's inequality.
>
> * **Reviewer**:  The results presented in section 3.3 and 3.4 are quite straight forward, and it is hard for me to see the values of those results based on the discussion in the current paper. More discussion and some potential applications of these results would be helpful. Otherwise, there is no reason to include some looser bounds in the main body of the paper.
>
>     **Answer**: These results are included to showcase the versatility that the Wasserstein distance bounds have. As mentioned to reviewer 61bD, the bounds based on the backward channel are included due to two main reasons:
>     1. The fact that this channel has been useful in the information theory community to characterize the rate--distortion function in the past. Therefore, we consider that this result may be useful to other people with that background, or others who may be capable to describe the backward channel for their particular application.
>
>     2. The fact that sometimes employing the geometric information of the sample space is more useful than employing the geometric information of the hypothesis space. For instance, imagine a scenario where the loss function $\ell(w,z)$ is not Lipschitz with respect to $w$, but it is Lipschitz with respect to $z$  (e.g., hinge, logistic, or Huber losses [24]) and assume also that the sample space has a bounded diameter. Then, even if you only know $P_{W|Z_i}$, you can reach a useful bound with the following steps:
>
>         1. realize that the backward channel bound in terms of the Wasserstein distance holds,
>
>         2. get the total variation bound as shown in Appendix A.2,
>
>         3. use the fact that the total variation is an $f$-divergence and therefore by Jensen's inequality and [Poliansky 2019, Prop. 7.1-4] you may bound that by the total variation between the joint distribution $P_{W,Z_i}$ and the product distribution $P_W \otimes P_{Z_i}$, and finally
>
>         4. characterize the bound with the total variation or any other $f$-divergence that you know how to compute by means of the joint-range strategy.
>
>     This second point is also related to why we included some examples of the obtainable bounds with the joint range strategy.
>     We wanted to showcase that these results open the floor to measure the stability of the algorithms with many different metrics, since sometimes one cannot prove analytically that the Wasserstein distance or the relative entropy are small, but maybe can do so for the Hellinger distance.
>     Then, using the bound from section 3.4, if $H(P_{W|Z_i},P_W) \in \mathcal{O}(1/\text{poly}(n))$, then $\overline{\text{gen}}(W,S) \in \mathcal{O}(1/\text{poly}(n))$.
>
>     ```Y. Poliansky. 1st lecture notes for the course "Information Theoretic Methods in Statistics and Computer Science" 2019-2020. ```
>
> * **Reviewer:** The comparison of the Wasserstein distance-based bounds presented in Appendix C should be included in the main body of the paper to demonstrate the tightness of the proposed bound. It would also be helpful to construct a table which summarizes the comparisons between all these bounds.
>
>     **Answer:** This is a great idea that will improve the reader experience, thank you. We now included an informal proposition right before Example 1 summarizing the relationships between the different Wasserstein distance bounds.
>
>     As for the table, we also think this is a great idea, but unfortunately due to page limitations we are unable to include that in the main text. As suggested by Reviewer SCNA, we will create a figure summarizing the comparisons between all these bounds, but it will have to be delegated to Appendix C.
>
> * **Reviewer**: The ICIMI bound from [12] could be added in Figure 1 as another baseline.
>
>     **Answer**: The bound from [12] is calculated with the squared $\ell_2$ norm and not the $\ell_2$ norm. In order to include this bound as a baseline, the conditional CGF using the squared $\ell_2$ norm should be calculated, which is a complicated task. Moreover, given that the bound from [12] has the same rate as [6] for the squared $\ell_2$ norm and the bound from [6] has the same rate for both the $\ell_2$ and squared $\ell_2$ norm, we anticipate that the order of such a bound will also be sub-optimal, that is $\mathcal{O}(1/\sqrt{n})$.

---

> > ### Comment · Reviewer_Bm5Z · 2021-08-31
> > **After rebuttal**
> >
> > Thanks for your clarification. My questions are addressed well, and the discussion of Gaussian location model is really helpful, which should be included in the paper.

---

> > > ### Author Response · Authors · 2021-09-02
> > > **Answer to after rebuttal comment**
> > >
> > > Dear Reviewer Bm5z,
> > >
> > > Thank you for your comment. Following your suggestion, we added a small commentary about the non-linearity of the individual mutual information bound (and how it leads to a sub-optimal bound in the Gaussian location model) at the end of page 5.

---

### Official Review · Reviewer_61bD · 2021-07-15

**Rating:** 6
**Confidence:** 3

**Summary:**

This paper considers the problem of obtaining generalization bounds using the statistical measure of dependence between the input and output of a learning algorithm. Tha main contribution of this paper is to provide a generalization bound using the Wasserstein distance between the posterior distribution and P(W). This result is further improved by considering the single sample case where the generalization bounds are based on the Wasserstein distance P(W|Z_i) and P(W). Also the authors extend their results to the setup of Steinke and Zakynthinou ‘20. The authors specifically follow the literature of information-theoretic generalization bounds, and compare their bounds to the existing bounds in the literature. In all cases, the authors show that the bounds using Wasserstein distance are tighter than the bound using mutual information (or KL).

Also, they show for a toy example of estimating the mean of gaussian distribution their bound provides “order-wise” improvement over the mutual information bounds where the mutual information based bounds gives upper bounds on the generalization that decays as 1/sqrt(n) but the Wasserstein distance based bound gives a bound that decays as 1/n.

The authors also use the joint-range technique to provide generalization bounds based on f-divergences.



**Ethical Concerns:**

There are no specific ethical concerns with this work.

**Limitations And Societal Impact:**

I think the limitations discussed by the authors in the paper is important. Also, I would like to add that the authors should add results regarding how we can choose a good metric and how we can find a good coupling.

**Main Review:**


-The Wasserstien distance is defined using a metric. The most important shortcoming I found with the paper is that there is no discussion on how we can find an appropriate metric for the hypothesis space. I just want to emphasize that the concept of using the geometry of the hypothesis space has been used in learning theory before. For instance for VC classes this seminal work considers the VC classes:

	Haussler, David. "Sphere packing numbers for subsets of the Boolean n-cube with bounded Vapnik-Chervonenkis dimension." Journal of Combinatorial Theory, Series A 69.2 (1995): 217-232.

Later it was used for obtaining bounds on the Radmacher complexity by chaining technique. Another question is what is the impact of the algorithm on the chosen metric? Assume we have a hypothesis space and two different algorithms for learning. Do we need to choose different metrics when analyzing different algorithms for the same hypothesis space?

-The concept of chaining mutual information bounds has been considered in [13], [11], and a recent work by Asadi, Abbe where the authors propose a learning algorithm based on their chaining method. I just want to understand what are the main differences of this paper compared to those results? The aforementioned papers  also provide bounds that depend on the learning algorithm as well as the geometry of the space.

-The example of the Gaussian location model implies that in some cases the proposed bounds can give much tighter characterization of the generalization bound.  Is there any specific practical problem that your generalization bound can give new insights and analysis? Also, does this generalization bound guide us to design a new training algorithm?

-Most of the results in the paper depends on the distribution of output conditioned on a single sample, i.e., P(W|Z_i). I think in most problems it should be hard to quantify this kernel. The authors also discuss the bounds that depend on P(W|S) however those bounds are much looser. Is there any way to improve the bounds based on P(W|S)?

-Another important property of the Wasserstein distance is that it is defined based on “inf” over the coupling. However, in the paper there is no discussion on how to find a good coupling? I think the authors should also comment on this aspect.

-Theorem 2: why do you want to add extra info R? I think we can add extra information by considering a specific coupling.

-Backward channel: I could not understand what is the main idea behind considering backward channel? I think characterizing the kernel P(S|W) is a very very hard problem. How can we use this bound in practice to gain a new insight? I would appreciate it if the authors comment on why they consider the backward channel in the paper. I have looked at the appendix on the connection to the rate distortion theory, but that section is not very clear.


Some typos:
What does the word “single letter” mean in the paper?
Line 33, there are three and in a row and it is difficult to understand the sentence.
Line 38, what is j?
Line 151: in the first sentence it seems there is a typo.



**Time Spent Reviewing:**

3

---

> ### Author Response · Authors · 2021-08-09
> **Answer to the comments [1/n]**
>
> Dear Reviewer 61bD,
>
> Thank you for reviewing our paper and for the detailed technical feedback. We will respond to your remarks (in order) in a chain of comments. We will first re-write your comments/questions and will follow them with our answers. Due to the detail needed to answer all the comments we might need more than one official comment. To assess readability, the answers will be titled "Answer to the comments [$i/n$]" for official comments $i = 1, \ldots, n$.
>
> * **Reviewer:** The Wasserstien distance is defined using a metric. The most important shortcoming I found with the paper is that there is no discussion on how we can find an appropriate metric for the hypothesis space. I just want to emphasize that the concept of using the geometry of the hypothesis space has been used in learning theory before. For instance for VC classes this seminal work considers the VC classes:
>
>     ```Haussler, David. "Sphere packing numbers for subsets of the Boolean n-cube with bounded Vapnik-Chervonenkis dimension." Journal of Combinatorial Theory, Series A 69.2 (1995): 217-232.```
>
>    Later it was used for obtaining bounds on the Radmacher complexity by chaining technique. Another question is what is the impact of the algorithm on the chosen metric? Assume we have a hypothesis space and two different algorithms for learning. Do we need to choose different metrics when analyzing different algorithms for the same hypothesis space?
>
>     **Answer:** Thank you for the reference. Just to be clear, we did not claim that studying the geometry of the space was not done before; in the introduction (lines 25--28), we mention how classical approaches studied the complexity and the geometry of the hypothesis space, referring to [2,3] as classic textbooks on the topic.
>
>     Regarding on how to choose the metric: any metric under which the loss function is Lipschitz is valid. This is a property of the hypothesis space (in case of the forward channel) and the loss function, and *not* of the algorithm. Therefore, if two different algorithms (that produce the same hypothesis space) are evaluated with the same loss function, then they can both be characterized with the same metric. Usually, the metric of choice becomes apparent based on the loss function.  As a simple example, consider samples of the type $z = (x,y)$:
>     * *Regression:* If a norm is used as the loss function $\ell(w,z) = \lVert w - y \rVert$, then such a norm is also a good choice for a metric since by the reverse triangle inequality the loss is 1-Lipschitz under that metric: $\big \lvert \lVert w - y \rVert - \lVert w' - y \rVert \big \rvert \leq \lVert w - w' \rVert$ for all $w, w' \in \mathcal{W}$.
>
>     * *Classification:* If the 0-1 loss is used as the loss function $\ell(w,z) = \text{Ind}(w \neq y)$, then the discrete metric is a good choice since the loss is also 1-Lipschitz under this metric: $\lvert \text{Ind}(w \neq y) - \text{Ind}(w' \neq y) \big \rvert \leq \text{Ind}(w \neq w')$.
>
>     With that said, it is possible that the loss function is Lipschitz with respect to multiple metrics, e.g., a bounded loss function represented as a norm is Lipschitz with respect to that norm and the discrete metric. In these situations the algorithm of choice can dictate which is the metric employed either because the analysis is simpler or because we can guarantee a better generalization bound with it.
>
>     In any case, the presented bounds are general enough so that they can adapt in different situations, and therefore there is not "a best way" to find the metric for a particular setting. For instance, consider a case where the $\ell_2$ norm is a clear choice. Then, it could be that a more knowledgeable person shows that the loss is also Lipschitz under a higher order norm and obtains a sharper bound that way.
>
> * **Reviewer**: The concept of chaining mutual information bounds has been considered in [13], [11], and a recent work by Asadi, Abbe where the authors propose a learning algorithm based on their chaining method. I just want to understand what are the main differences of this paper compared to those results? The aforementioned papers also provide bounds that depend on the learning algorithm as well as the geometry of the space.
>
>     **Answer**: In [13], the authors follow the chaining technique from Dudley [R. van Handel 2014, Th. 5.24] together with [4, Th. 1] to bound separable, subgaussian processes.
>
>     * The first difference between the two theoretical results is their requirements (or assumptions):
>
>         * In [13] it is required that $\lbrace \text{gen}(w,S) \rbrace_{w \in \mathcal{W}}$ is a separable subgaussian process under the metric $\rho$. This means that the following inequality must hold for all $w, w' \in \mathcal{W}$ and all $\lambda \in \mathbb{R}$:
>         $$ \log \mathbb{E} \Big[ e^{\lambda \big( \text{gen}(w,S) - \text{gen}(w',S)\big)} \Big] \leq \frac{\lambda^2 \rho( \text{gen}(w,S),\text{gen}(w',S))^2}{2}.$$
>         This can be interpreted as an "in probability" requirement of Lipschitzness of the generalization error.
>
>         * In this paper, for many of the results, we require that the loss function $\ell(\cdot,z)$ is Lipschitz for all $z \in \mathcal{Z}$ under the metric $\rho$. This assumption is more strict since, if $\ell(\cdot,z)$ is Lipschitz, then $\lbrace \text{gen}(w,S) \rbrace_{w \in \mathcal{W}}$ is subgaussian, but the reciprocal is not necessarily true.
>
>     * The second difference is in the bound itself:
>
>         * In [13] several refined quantizations $W_k$ of the hypothesis random object $W$ are calculated to bound the generalization error by a weighted average of a function of the mutual information between these quantizations $W_k$ and the dataset $S$. There the metric is used to describe how the hypothesis space is partitioned to obtain such quantization rules. This is interesting because the quantization avoids the problem of potentially infinite mutual information.
>
>         * We instead look at the minimum cost to go from the distribution of the hypothesis after observing some samples to its marginal distribution, e.g., $\mathbb{W}(P_{W|Z_i}, P_W)$. Here the metric is used to quantify how far away are, on average, the realizations of both probability distributions if they are coupled in the best way possible.
>
>     We do not have any general direct comparison between the bounds of [13] and ours as of now. Nonetheless, our bounds have the flexibility that allow us to consider either the hypothesis or the sample space (backward channel).  For instance, for the main example of that paper, i.e. [13, Example 1], our bounds using the backward channel are tighter than the bounds arising from the chaining technique:
>
>     * In this example they consider the canonical Gaussian process. The hypothesis space is $\mathcal{W} = \lbrace w \in \mathbb{R}^2: \lVert w \rVert_2 = 1 \rbrace$, the samples are $Z_i \sim \mathcal{N}(0,I_2)$, the loss function is $\ell(w,z) = - w^T z$, and the hypothesis is selected with the ERM algorithm, i.e., $w^\star = \text{arg} \min_{w \in \mathcal{W}}  \big \lbrace \frac{1}{n} \sum_{i=1}^n \ell(w,z_i) \big \rbrace $.
>
>     * In this setting, by Cauchy-Schwarz we see that the loss $\ell(w,\cdot)$ is 1-Lipschitz for all $w \in \mathcal{W}$, i.e. $| - w^Tz + w^Tz'| \leq \lVert w \rVert_2 \lVert z - z' \rVert_2 \leq  \lVert z - z' \rVert_2$ for all $z, z' \in \mathcal{Z}$. Therefore, the backward channel equivalent of Theorem 1 holds.
>
>     * Moreover, the function is 1-subgaussian under $P_Z$ for all $w \in \mathcal{W}$. Therefore, as shown in Appendix A.1 our bound is tighter than [6], which is shown to be tighter than [13] in this setting (see [6, Sec. IV-B, in particular Figs. 1 and 2]).
>
>     ```Ramon Van Handel. "Probability in high dimension." Princeton University, 2014.```

---

> ### Author Response · Authors · 2021-08-09
> **Answer to the comments [2/n]**
>
> This is a continuation of the answer titled "Answer to the comments [1/n]".
>
> * **Reviewer:** The example of the Gaussian location model implies that in some cases the proposed bounds can give much tighter characterization of the generalization bound. Is there any specific practical problem that your generalization bound can give new insights and analysis? Also, does this generalization bound guide us to design a new training algorithm?
>
>     **Answer:**
>
>     * As mentioned to reviewer EsUW, the new bounds in terms of the relative entropy directly improve upon previous results. For instance, the incorporation of the BH inequality in Cor. 1 and Cor. 2 improves [6, Prop.1] and [7, Th. 2.5], respectively, and pulling the expectation with respect to the samples outside of the square root in Cor. 1 improves [6, Prop. 1] by Jensen's gap. Therefore, one can follow the analyses for the different algorithms in these papers and readily strengthen the bounds this way. Note that Jensen's gap can be large in certain cases, e.g., when the variance of the relative entropy values for different input instances or different subsets of the data is large.
>
>     * In [14], the authors show that the hypothesis learned with ERM and the bound $\mathbb{W}(P_{W|S=s},P_W)$ as a regularization term and "forcing" the marginal distribution of the hypothesis $P_W$ to be the delta distribution at the origin is equivalent to the common minimum description length problem or Occam's razor.
>
>         This is great news, since it strengthens our prior beliefs from learning theory. This connection is still true if one uses as a regularizer our bound based on individual samples $\mathbb{W}(P_{W|Z_i=z_i},P_W)$ instead, which is tighter and thus further strengthens those beliefs.
>
>     * Also, these results reinforce the intuition that more stable algorithms generalize better. Having such a result in terms of the Wasserstein distance is good because it indicates that such an intuition is not rooted in the way we measured the stability (i.e., how close the distribution of the hypothesis is before and after observing the data), since the Wasserstein distance looks at the*minimum* cost to go from one distribution to the other. Moreover due to the versatility of the Wasserstein distance to produce upper bounds in different metrics (see Section 3.4), it allows us to understand and measure the algorithm's stability in different ways.
>
> * **Reviewer:** Most of the results in the paper depends on the distribution of output conditioned on a single sample, i.e., $P(W|Z_i)$. I think in most problems it should be hard to quantify this kernel. The authors also discuss the bounds that depend on $P(W|S)$ however those bounds are much looser. Is there any way to improve the bounds based on $P(W|S)$?
>
>     **Answer:** In Th. 1 and Th. 2 of the paper, we present results that depend on $P_{W|Z_i}$ and $P_{W|S^{-i}}$ respectively.
>     Then we show that those results are tighter than (or tighter than 2 times) those based on $P_{W|S}$, see Appendix C.1.
>     For instance, in the Gaussian location model example, they are order-wise tighter.
>     So, in a sense, one could say that these two refinements are a way to improve the bounds that depend on $P_{W|S}$.
>     At the moment, we are not aware of any other possible improvements to previous bounds based on $P_{W|S}$.
>
>     In terms of practicality, the bounds depending on $P_{W|S^{-i}}$ can be obtained with a similar degree of difficulty than those based on $P_{W|S}$, e.g., see [7], and in some situations, considering $P_{W|Z_i}$ can actually make the computation of the bound simpler, e.g., see [6].
>
> * **Reviewer:** Another important property of the Wasserstein distance is that it is defined based on “inf” over the coupling. However, in the paper there is no discussion on how to find a good coupling? I think the authors should also comment on this aspect.
>
>     As we mentioned before for the choice of a metric, the objective of this paper is to provide a general recipe to obtain tighter characterizations of the generalization error. The provided recipe is general and flexible and, as such, it comes with the pitfall that it is not as simple to use as a plug-and-play formula.
>
>     Once an appropriate metric is chosen and we know the generalization error is bounded from above by the Wasserstein distance, how to obtain an analytical bound or an estimation for that quantity is a case-by-case matter and outside of the scope of what we intended with this paper.
>
>     Also, in the limitations of the paper we mention how currently estimating the Wasserstein distance is a complicated matter.
>     However, we hope that the realization that this kind of bounds can match the complexity of the real generalization error (Example 1) serves as motivation for further research in the theory of Wasserstein distance characterization and estimation.
>     For example, we consider that works like [Rowland 2019, Chizat 2020, and Staerman 2022] show promise in this direction.
>     In addition, the projected Wasserstein distance seems to be a simpler to estimate and characterize metric; this metric can still be used to characterize the generalization error since it bounds from above the standard Wasserstein distance [Rowland 2019, Prop.\ 3.4].
>
>     ```Mark Rowland, Jiri Hron, Yunhao Tang, Krzysztof Choromanski, Tamas Sarlos, and Adrian Weller, "Orthogonal Estimation of Wasserstein Distances", AISTATS 2021```
>
>     ```Lénaïc Chizat, Pierre Roussillon, Flavien Léger, François-Xavier Vialard, and Gabriel Peyré, "Faster Wasserstein Distance Estimation with the Sinkhorn Divergence", NeurIPS 2020```
>
>     ```Guillaume Staerman, Pierre Laforgue, Pavlo Mozharovskyi, and Florence d’Alché-Buc, "When OT meets MoM: Robust estimation of Wasserstein Distance", AISTATS 2021```
>
> * **Reviewer:** Theorem 2: why do you want to add extra info R? I think we can add extra information by considering a specific coupling.
>
>     **Answer:** The theorem holds even without the extra random object $R$, e.g., let $R \sim \delta_0$.
>     We will make that explicit with a short sentence after the statement of the result.
>
>     As mentioned after Corollary 2, the reason why we include $R$ is that it facilitates the incorporation of some knowledge necessary to characterize the hypothesis distribution in certain problems.
>     For instance, in [7, 9], $R$ represents the batch indices of SGLD, which allows us to get a closed-form expression of the conditional distribution of the hypothesis and develop tighter bounds.
>
>     Moreover, expressing the result this way, it is easier to see how the presented result is tighter than [7, Th. 2.5].
>     Connecting with some of the previous points, Cor. 2 in its relative entropy form directly improves upon [7, Th. 2.5] and generates a practical bound.
>
> * **Reviewer:** Backward channel: I could not understand what is the main idea behind considering backward channel? I think characterizing the kernel P(S|W) is a very very hard problem. How can we use this bound in practice to gain a new insight? I would appreciate it if the authors comment on why they consider the backward channel in the paper. I have looked at the Appendix on the connection to the rate distortion theory, but that section is not very clear.
>
>     **Answer:** There are two main reasons for which we include the results based on the backward channel:
>
>     1. Although we agree with the reviewer that the backward channel is difficult to characterize in many settings, we appreciate how considering this channel has been useful for the information theory community in the past.
>     To name a few examples, this approach has helped:
>
>         * to derive an analytical solution to the rate–distortion function for a particular input signal distribution, e.g., Bernoulli or Gaussian [18, Ch. 27.1];
>
>         * to derive an analytical bound for certain distribution families, e.g., bounded variance; or,
>
>         * to derive an analytical bound for certain distortion families, e.g., difference, additive, or autoregressive distortions [36, Secs. 4.3, 4.6, and 4.7].
>
>     Therefore, we considered that stating such a result could be useful for people from an information theory (or related) background who may be capable of describing the backward channel for their particular application.
>
>     2. Sometimes employing the geometric information of the sample space is more useful than employing the geometric information of the hypothesis space.
>     For instance, imagine a scenario where the loss function $\ell(w,z)$ is not Lipschitz with respect to $w$, but it is Lipschitz with respect to $z$, and assume also that the sample space has a bounded diameter.  Then, even if you only know $P_{W|Z_i}$, you can reach a useful bound with the following steps:
>
>         1. realize that the backward channel bound in terms of the Wasserstein distance holds,
>
>         2. get the total variation bound as shown in Appendix A.2,
>
>         3. use the fact that the total variation is an $f$-divergence and therefore by Jensen's inequality and [Poliansky 2019, Prop. 7.1-4] you may bound that by the total variation between the joint distribution $P_{W,Z_i}$ and the product distribution $P_W \otimes P_{Z_i}$, and finally
>
>         4. characterize the bound with the total variation or any other $f$-divergence that you know how to compute by means of the joint-range strategy.
>
>     ```Y. Poliansky. 1st lecture notes for the course "Information Theoretic Methods in Statistics and Computer Science" 2019-2020.```

---

> ### Author Response · Authors · 2021-08-09
> **Answer to the comments [3/n], n = 3**
>
> This is a continuation of the answer titled "Answer to the comments [2/n]".
>
> * **Reviewer**: What does the word “single letter” mean in the paper?
>
>     **Answer**: When we say "single-letter" we use the terminology from information theory where a problem that depended on $n$ random variables is reduced to $n$ problems that depend on one random variable.
>     For instance, instead of considering $\mathbb{W}(P_{W|Z_1, \ldots, Z_n},P_W)$ we now consider $\frac{1}{n} \sum_{i=1}^n \mathbb{W}(P_{W|Z_i}, P_W)$, and in the case where $P_{W|Z_i} = P_{W|Z_1}$ for all $i$ this reduces to a single simpler problem $\mathbb{W}(P_{W|Z_1}, P_W)$. I believe in other disciplines this is also known as tensorization (even though I am not sure if that term only applies for the case where a function applied to the product distribution is the same to the product of that function applied to each of the marginal distributions).
>
> * **Reviewer**: Line 33, there are three and in a row and it is difficult to understand the sentence.
>
>     **Answer:** In line 33, to simplify the reading we just reverserd the order of the two sentences, that is "$[\cdots]$ which occurs for example when $W$ and $S$ are separately continuous and $W$ is a deterministic function of $S$".
>
> * **Reviewer**: Line 38, what is j?
>
>     **Answer**: In line 38, $j$ is the set of indices that decides which samples are not included in the conditioning. We employed the same notation than later in the rest of the paper (e.g. Th. 2). We did not mention that to avoid unnecessary notation in the introduction. To help the comprehension, we modified the sentence as follows "$[\cdots]$ random subsets of the data, i.e., $D_{\text{KL}} \big(P_{W|s} \lVert P_{W|s_{j^c}} \big)$ where $j \subseteq [n]$".
>
> * **Reviewer**:  Line 151: in the first sentence it seems there is a typo.
>
>     **Answer:** The text from line 151 is now removed, following the advice from reviewers Bm5Z and SCNA we included an informal proposition summarizing the relationships between the different Wasserstein distances. Preceding this proposition, there is a short paragraph describing the key points to obtain such results, which is an equivalent to what was in line 151: "It is possible to prove that Theorem 1 is tighter than [14, Theorem 1]. This results by studying the KR dual representation of the Wasserstein distance and noting that the conditional distribution $P_{W|Z_i}$ is a smoothed version of the forward channel, i.e., $P_{W|Z_i} = \mathbb{E}[P_{W|S}|Z_i]$. Comparisons with Theorem~2 are also possible using similar arguments and the triangle inequality. These results are informally summarized below and presented with more details and the proofs in Appendix C.1.".
>
>
> **END:** That is all, thank you again for the patience of reading through such a long thread of answers.

---

> ### Comment · Reviewer_61bD · 2021-08-15
> **POST-REBUTTAL**
>
> Dear Authors,
>
> I would like to thank the authors for their detailed responses. However, if accepted, the authors should very carefully address the following:
>
> 1) Chaining method for MI bounds and PAC-Bayes bounds: As discussed in the response letter, there are results providing generalization bounds that are both considering the geometry and the learning algorithm. I think the authors should clearly discuss the differences between their approach and the existing results. Also, the authors should cite [a,b].
> 2) How to find a metric to obtain a tightest possible bound: I found the response by the authors interesting, and they should include the discussion in the paper.
> 3) How to find coupling so that the bound can be evaluated: My main concern is that finding a "good" coupling is too cumbersome so it limits the applications of the bounds to toy examples. Nonetheless, I think the results in this paper can be a starting point to address this issue in case-by-case scenarios.
>
> Specifically, I think it would be nice if the authors provide detailed discussion in the paper for the aforementioned points. I increased my score to 6.
>
>
> [a]J. Audibert and O. Bousquet. PAC-Bayesian generic chaining. In Advances in Neural Information Processing Systems , pages 1125–1132, 2004
> [b] J. Audibert and O. Bousquet. Combining PAC-Bayesian and generic chaining bounds. Journal of Machine Learning Research, 8(Apr):863–889, 2007.

---

> > ### Author Response · Authors · 2021-08-30
> > **Answer to post-rebuttal**
> >
> > Dear Reviewer 61bD,
> >
> > Thank you for your answer and increasing the score. Based on your suggestions we now have made the following modifications to the paper:
> >
> > * We created an appendix section (now Appendix A) where we discuss the evolution of geometry based bounds based on random processes going through the Lipschitz maximal inequality [R. van Dalen, Lemma 5.7], to the original chaining technique [R. van Dalen, Theorem 5.24], to some PAC-Bayesian bounds [a,b], to conclude in the inclusion of the technique from [4, Lemma 1] in the chaining-based bounds from [13].
> >
> >   The appendix section continues comparing these bounds with the Wasserstein distance bounds in a similar fashion to what we did in our response to you.
> >
> >   The appendix section has a final subsection where we do a discussion on how to choose the metric and we give some examples. This discussion is a slight extension of our answer to your previous question.
> >
> >   This appendix section is referenced in the introduction when [13] is initially introduced and in section 3.5, where we point to the discussion on the choice of the metric.
> >
> > * Papers [a,b] are also referred to in the discussion, more precisely in the PAC-Bayes bounds section of the discussion. There, they are referred as an example of PAC-Bayes bounds based on the hypothesis' geometry.
> >
> > * Regarding how to find a coupling, we did not make any modifications. As discussed before, the objective of the paper was to provide a flexible and general formula, and the study of how to find optimal couplings (while interesting and a path for future research) is out of the scope of the presented document.
> >
> > Correction: We had a typo in the answer we gave you when writing the formula denoting that the process $\lbrace \text{gen}(w,S) \rbrace_{w \in \mathcal{W}}$ is sub-gaussian under a metric $\rho$. As you already know, the correct condition than must be held for all $w, w' \in \mathcal{W}$ and all $\lambda \in \mathbb{R}$ is
> > \begin{equation*}
> > \log \mathbb{E} \Big[e^{\lambda\big(\text{gen}(w,S)-\text{gen}(w',S)\big)} \Big] \leq \frac{\lambda^2 \rho\big(w,w'\big)^2}{2}.
> > \end{equation*}
> > This typo does not change the content of our previous answer, but we thought it would be nice to state the correct equation for a better context.

---

### Official Review · Reviewer_EsUW · 2021-07-17

**Rating:** 7
**Confidence:** 2

**Summary:**

This work provides novel generalization bounds, which limit (in expectation) the gap between a classifier’s empirical risk, i.e. how well it performs on the training data, and population risk, i.e. how well it performs on the true distribution. Generalization bounds can be in terms of many parameters of the problem, and this paper focuses on Wasserstein distances. Specifically, it provides bounds in terms of the Wasserstein distance between the hypothesis distribution before and after a single sample (“single-letter”), before and after a random subset of the samples (“random-subset”), and before and after the full set of samples (“full-dataset”). These Wasserstein distances were studied in previous work as well, but these bounds are tighter than the predecessor bounds.


**Limitations And Societal Impact:**

The paper itself admits “The Wasserstein distance is difficult to characterize and/or estimate,” and the one application presented (Gaussian mean estimation) is already on the easy side because its generalization can directly be computed. Is there an application where these results produce the generalization bounds, even compared to non-information based techniques?

**Main Review:**

The first four main results of the paper are for single-letter and random-subset generalization bounds in the standard and randomized-subsample settings. Note that in the latter setting, twice the typical number of samples are drawn, but coin flips are used to decide which subset of those samples is actually used; the distributions considered differ in whether the results of those coin flips / the identities of the samples are known. The bounds in the standard setting are showcased by considering a particular application regarding estimating the mean of a Gaussian distribution given its covariance matrix and comparing the proposed bounds to previous Wasserstein / mutual information bounds. Since this setting is simple, the precise generalization error can be computed and is plotted as well; the new bounds are much closer to the correct generalization error than previous work.

The paper then also discusses how the results above can be made to work with backward channels instead of forward channels (e.g. how knowing the hypothesis changes the distribution over samples), as well as how they can be converted to other information measures, such as launtum information.

Overall, I think this is a thorough collection of theory results that pushes the boundaries relative to a variety of prior work. I am a bit concerned about the practicality of these results (see the Limitations and Societal Impact section), but I would still lean towards acceptance.

Miscellaneous:
- Line 3: “and their analogous in” analogous -> analogues.
- Line 153: “tighter than (1)” Took a moment for me to parse this reference, I think it would be clearer if it were more explicit e.g. Inequality (1) or perhaps Formula (1)? Similarly on line 160.
- Line 210: “impact on the samples’ identities” on -> of

-----
I have read the author's response and they have addressed my concern about there being cases where these bounds improve on practical cases, so I have adjusted my score accordingly to an accept.

**Time Spent Reviewing:**

3 hours

---

> ### Author Response · Authors · 2021-08-09
> **About the characterization of the Wasserstein distance and the application of these results**
>
> Dear Reviewer EsUW,
>
> Thank you for reviewing and assessing our paper. Please, let us answer to your comments below.
>
> Regarding the limitations and societal impact:
>
> As noted, the Wasserstein distance is complicated to characterize and/or estimate. Nonetheless, there are two points we want to note in favor of the presented results in terms of their practicality:
>
> 1. The bounds based on the Wasserstein distance allow us to generate new bounds in terms of the easier to estimate relative entropy, which already improve upon previous results. For instance, there are two ways in which the proposed bounds can tighten previous ones:
>     * the BH inequality prevents that the larger relative entropy values make the bound vacuous. Therefore, Cor.  1 improves upon  [6, Prop.  1] and Cor.  2 improves upon [7, Th. 2.5], and
>     * it pulls the expectation with respect to the samples, e.g., $P_{Z_i}$, outside of the square root, thus improving the bound due to Jensen's gap. Hence, this way Cor.  1 further improves upon [6, Prop.  1]. Note that, in certain cases, this gap grows as $\Theta(\sigma_\alpha^\alpha)$, where $\sigma_\alpha$ is the $\alpha$ absolute central moment [Gao 2019]. Therefore, the current practical bounds on logistic regression, Gaussian Processes, and SGLD (assuming a bounded loss) from [6] are readily tightened.
>
>     ```Xiang Gao, Meera Sitharam, Adrian E. Roitberg, "Bounds on the Jensen Gap, and implications for mean-concentrated distributions", AJMAA```
>
> 2. In this paper, we show evidence (see Example 1) that the characterization of the generalization error via Wasserstein distance can be tight. We hope that this realization can serve as motivation for further research in the theory of Wasserstein distance characterization and estimation.
>
>     We are particularly hopeful due to recent results on Wasserstein distance estimation [Rowland 2019, Chizat 2020, and Staerman 2022]. Moreover, the projected Wasserstein distance seems to be a simpler to estimate and characterize metric; this metric can also be used to characterize the generalization error since it bounds from above the standard Wasserstein distance [Rowland 2019, Prop. 3.4].
>
>     ```Mark Rowland, Jiri Hron, Yunhao Tang, Krzysztof Choromanski, Tamas Sarlos, and Adrian Weller, "Orthogonal Estimation of Wasserstein Distances", AISTATS 2021```
>
>     ```Lénaïc Chizat, Pierre Roussillon, Flavien Léger, François-Xavier Vialard, and Gabriel Peyré, "Faster Wasserstein Distance Estimation with the Sinkhorn Divergence", NeurIPS 2020```
>
>     ```Guillaume Staerman, Pierre Laforgue, Pavlo Mozharovskyi, and Florence d’Alché-Buc, "When OT meets MoM: Robust estimation of Wasserstein Distance", AISTATS 2021```
>
> Regarding the miscellaneous:
>
> - We fixed the mistakes from lines 3 and 210.
> - For the IEEE style of referring equations from line 153, we included now a light blue color to the hyperlinks so it is easier to navigate.

---

### Decision · Program_Chairs · 2021-09-27

**Decision:**

Accept (Poster)

**Comment:**

Overall, the reviewers were positive about the paper. My main concern is regarding its novelty: it seems that this paper combines Wang et al. 2019 with the individual sample approach proposed by Bu et al., 2020 and with the conditional mutual information approach proposed by Steinke and Zakynthinou, 2020, respectively. There are also limitations in terms of how useful the Wasserstein-based generalization bounds proposed in this paper are in practice, particularly regarding estimation of Wasserstein distance from data. The bounds seem difficult to instantiate even for very basic examples. As noted by several reviewers, it is unclear how the paper's results would translate to more complex models. Note that recent work in the area of information-theoretic generalization bounds provide numerical results to fairly sophisticated settings (e.g., the analysis of SGLD by Negrea and Haghifam et. al.) and, despite the bounds themselves often being vacuous, at least they correlate with the generalization behavior of neural networks on real-world data. The reviewers also pointed out several changes to the paper that I view as "mandatory," especially the points raised by reviewer 61bD.

However, I do find that this paper moves the needle on generalization bounds forward. Despite mostly relying on a combination of existing theoretical tools used in the area of information-theoretic generalization bounds, it combines these tools in a novel way. The limitations/future work section is particularly thoughtful and reveals interesting new research directions. Together, the positive aspects of the paper outweigh its limitations, making me lean towards acceptance --- with the caveat that the authors should add the changes promised in their rebuttal.